# BrainUICL: An Unsupervised Individual Continual Learning Framework for EEG Applications

**Yangxuan Zhou**[1,2], **Sha Zhao**[1,2]*, **Jiquan Wang**[1,2], **Haiteng Jiang**[3,4,1], **Shijian Li**[1,2],
**Tao Li**[3,4,1], **Gang Pan**[1,2,4]

[1]State Key Laboratory of Brain-machine Intelligence, Zhejiang University
[2]College of Computer Science and Technology, Zhejiang University
[3]Department of Neurobiology, Affiliated Mental Health Center & Hangzhou
Seventh People's Hospital, Zhejiang University School of Medicine
[4]MOE Frontier Science Center for Brain Science and Brain-machine Integration,
Zhejiang University
{zyangxuan, szhao, wangjiquan, h.jiang}@zju.edu.cn;
{shijianli, litaozjusc, gpan}@zju.edu.cn;

## Abstract

Electroencephalography (EEG) is a non-invasive brain-computer interface technology used for recording brain electrical activity. It plays an important role in human life and has been widely uesd in real life, including sleep staging, emotion recognition, and motor imagery. However, existing EEG-related models cannot be well applied in practice, especially in clinical settings, where new patients with individual discrepancies appear every day. Such EEG-based model trained on fixed datasets cannot generalize well to the continual flow of numerous unseen subjects in real-world scenarios. This limitation can be addressed through continual learning (CL), wherein the CL model can continuously learn and advance over time. Inspired by CL, we introduce a novel Unsupervised Individual Continual Learning paradigm for handling this issue in practice. We propose the BrainUICL framework, which enables the EEG-based model to continuously adapt to the incoming new subjects. Simultaneously, BrainUICL helps the model absorb new knowledge during each adaptation, thereby advancing its generalization ability for all unseen subjects. The effectiveness of the proposed BrainUICL has been evaluated on three different mainstream EEG tasks. The BrainUICL can effectively balance both the plasticity and stability during CL, achieving better plasticity on new individuals and better stability across all the unseen individuals, which holds significance in a practical setting. The source code is available at https://github.com/xiaobaben/BrainUICL.

## 1 Introduction

Electroencephalography (EEG) is a non-invasive brain-computer interface (BCI) technology, recording brain electrical activity through electrodes placed on the scalp. Due to the non-invasive nature and relatively high temporal resolution, EEG plays an important role in human life and has been widely used in practice, especially in clinical settings (i.e., sleep staging (Perslev et al., 2019; Aboalayon et al., 2016), emotion recognition (Song et al., 2018; Cowie et al., 2001), motor imagery (Tabar & Halici, 2016) and disease diagnosis (Petit et al., 2004; Jeong, 2004)). However, existing EEG-related models cannot perform well in real life. In practical situations, there are gradually varying new subjects every day. Moreover, there are significant **individual discrepancies** (i.e., physiological structures, physical characteristics) among different subjects. Such EEG-based models trained on fixed datasets cannot generalize well to the new unseen individuals. The above limitation motivates us to address this issue for practical applications. Fortunately, this problem can be reduced to **continual learning** (CL),

---

*Corresponding authors.

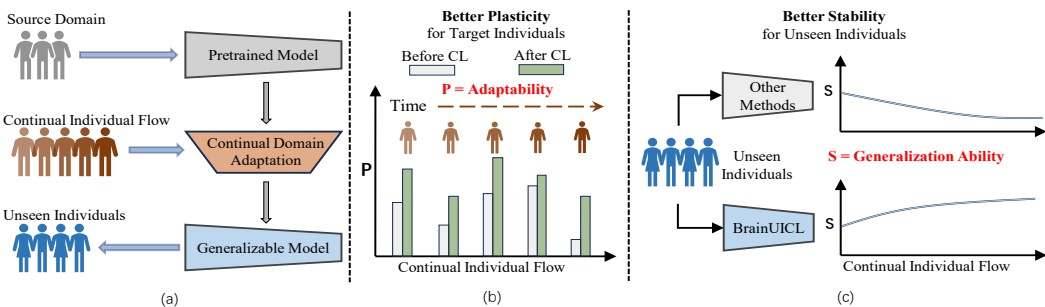

Figure 1: (a). The proposed paradigm of Unsupervised Individual Continual Learning; (b). Continual domain adaptation for better Plasticity; (c). Generalizable model for better Stability.

wherein the CL model can learn and advance by continuously absorbing new knowledge. The major challenge in CL is to overcome the Stability-Plasticity (SP) dilemma (Mermillod et al., 2013), with **Plasticity (P)** denotes the model's adapting ability to newly emerging individuals, while **Stability (S)** indicates the model's generalization ability to both previously seen and unseen individuals (i.e., new subjects). Inspired by CL, and considering that the incoming individuals lack ground truth labels, we propose a novel **U**nsupervised **I**ndividual **C**ontinual **L**earning (**UICL**) paradigm for handling EEG tasks in practical applications shown in Fig. 1 (a). Notably, considering the individual discrepancies, we treat each subject as a distinct individual domain (Yang et al., 2023) in our study. As shown in Fig. 1 (a), the pre-trained model is required to continuously adapt to multiple individual target domains one by one while absorbing the new knowledge to advance, and finally becomes a universal expert for all unseen individuals. Our main objectives are twofold: *(1) Better Plasticity: the model can adapt well to each new subject from the continual individual flow shown in Fig. 1 (b). (2) Better Stability: the model can achieve stronger generalization ability on all the unseen subjects after continuously learning the knowledge from the continual individual flow shown in Fig. 1 (c).*

However, it is not a straightforward task to enable the model to continuously adapt well to multiple newly emerging subjects (better P) and simultaneously improve its generalization ability for all unseen subjects (better S). There are three main reasons. First, a better Plasticity is difficult to obtain, because the individual discrepancies among the continual individual flow lead to continual domain shifts between the distribution of the source domain and that of the individual target domains. Second, the Stability could decrease on all unseen individuals, because the model may be overfitted to some individual target domains for a better plasticity. What's worse is that if the model adapts to some outlier individuals, the model may dramatically degrade in performance and may not be able to recover during subsequent continual adaptation (Wang et al., 2022). Third, balancing the plasticity and stability is challenging, which means the model needs to ensure its adaptability to new individuals while improving the generalization ability on all the unseen individuals. There have been some existing studies addressing similar issues, but they are not so adaptable in practice. For example, the studies Wang et al. (2022); Taufique et al. (2021; 2022) face a small quantity of varying target domains, and tackle the SP dilemma in scenarios such as object detection and image classification. They typically assume that the domain change in continual batches is minimal and conduct study at the sample level. However, the practical scenario is quite different, where there is a continual flow of numerous new subjects and there exist significant individual domain changes. Meanwhile, in real life, it is required to be conducted at the individual level (i.e., testing the EEG data of only one person at a time) instead of the sample level.

To achieve both better plasticity and stability, we propose a novel EEG-based **U**nsupervised **I**ndividual **C**ontinual **L**earning framework, called **BrainUICL**. It is well-suited to real-world scenarios where a large number of unseen and unordered individuals continuously emerge, enabling the model to continuously adapt to a long-term individual flow in a plug-and-play manner, while also balancing the SP dilemma during such CL process. We have designed two novel modules: the Dynamic Confident Buffer (DCB) and Cross Epoch Alignment (CEA) to tackle the aforementioned challenges. Specifically, the DCB employs a selective replay strategy that ensures the accuracy of labels for replay samples in an unsupervised setting while maintaining the diversity of these samples. The CEA module innovatively aligns the incremental model across different time states to prevent overfitting,

ensuring that the incremental model remains unaffected by varying learning trajectories, which is particularly relevant given that continual flows are unordered in real-world scenarios. Besides, it is worth pointing that BrainUICL is easy to be implemented without any modifications to the model structure. The contributions of this paper can be summarized as follows:

- We first explore the concept of the Unsupervised Individual Continual Learning(UICL) in EEG-related applications, **which is well-suited to the real-world scenario and** meets the practical needs in real life. The proposed BrainUICL framework can **effectively balance Stability-Plasticity dilemma during the CL process**.
- **We design novel DCB and CEA modules** to dynamically adjust the adaptation process during the long-term individual continual learning, overcoming the challenges of overfitting and preserving the knowledge learned from the past individual flow.
- Validated on **three different mainstream EEG tasks**, BrainUICL enables the model to adapt well to continual individual flow (**better Plasticity**), and achieve stronger generalization ability on all unseen individuals(**better Stability**), resulting in a win-win gain.

## 2 METHODOLOGY

### 2.1 PROBLEM SETUP AND PRELIMINARIES

Facing practical applications, we try to make the model not only adapt well to continuously incoming new subjects, but also generalize well to all the unseen subjects, taking advantage of the idea of unsupervised individual continual learning. We consider each subject as an individual domain. Formally, given multiple labeled individual domains (i.e., source domain, training set) $\mathcal{D}_S=\{\mathcal{X}_S^i, \mathcal{Y}_S^i\}_{i=1}^{\mathcal{N}_S}$ with $\mathcal{N}_S$ subjects, multiple unlabeled individual target domains (i.e., continual individual flow, incremental set) $\mathcal{D}_{\mathcal{T}}=\{\mathcal{X}_{\mathcal{T}}^i\}_{i=1}^{\mathcal{N}_{\mathcal{T}}}$ with $\mathcal{N}_{\mathcal{T}}$ subjects, and multiple labeled test domains (i.e., generalization set) $\mathcal{D}_G=\{\mathcal{X}_G^i, \mathcal{Y}_G^i\}_{i=1}^{\mathcal{N}_G}$ with $\mathcal{N}_G$ subjects, where $\mathcal{N}_G < \mathcal{N}_S \ll \mathcal{N}_{\mathcal{T}}$. Different individual target domains follow non-identical data distributions $\mathcal{P}(\mathcal{D}_{\mathcal{T}}^i) \neq \mathcal{P}(\mathcal{D}_{\mathcal{T}}^j)$, where $1 \leq i \neq j \leq \mathcal{N}_{\mathcal{T}}$. We denote the incremental model as $\mathcal{M}$ and its probability distribution as $\mathcal{P}(\mathcal{M})$, where $\mathcal{M}_0$ denotes the initial model trained from the source domain $\mathcal{D}_S$, and $\mathcal{M}_i$ denotes the current updated model when it has adapted to $\mathcal{D}_{\mathcal{T}}^i$. In our UICL setting, we consider the incremental model $\mathcal{M}$ is available with only an individual target domain at once. When $\mathcal{M}_i \to \mathcal{M}_{i+1}$ after each round updating, the corresponding distribution change can be described as $\Delta_{\mathcal{P}} = \Delta(\mathcal{P}(\mathcal{M}_i), \mathcal{P}(\mathcal{M}_{i+1}))$. During the CL process, the BrainUICL will gradually increase the penalty on the incremental individual target domain with continual update iterations, leading to smaller distribution change, i.e., $\lim_{i \to +\infty} \Delta_{\mathcal{P}} = 0$. The objective of BrainUICL is to enable the incremental model $\mathcal{M}$, trained from a small source domain $\mathcal{D}_S$, to adapt to multiple individual target domains $\mathcal{D}_{\mathcal{T}}$ and improve the generalization ability for the unseen test domain $\mathcal{D}_G$ after continuously absorbing new knowledge. During each round iteration, our goal can be described as follows:

$$\min_{\theta_M}(\mathbb{E}_{(\mathcal{X}_{\mathcal{T}}^i, \mathcal{Y}_{\mathcal{T}}^i) \sim \mathcal{D}_{\mathcal{T}}^i} \mathcal{L}(\mathcal{M}(\mathcal{X}_{\mathcal{T}}^i), \mathcal{Y}_{\mathcal{T}}^i) + \mathbb{E}_{(\mathcal{X}_G, \mathcal{Y}_G) \sim \mathcal{D}_G} \mathcal{L}(\mathcal{M}(\mathcal{X}_G), \mathcal{Y}_G)) \tag{1}$$

where $\mathcal{M}_i$ parameterized by $\theta_{\mathcal{M}_i}$. Here, $\mathbb{E}_{(\mathcal{X}_G, \mathcal{Y}_G) \sim \mathcal{D}_G} \mathcal{L}(\mathcal{M}(\mathcal{X}_G), \mathcal{Y}_G)$ can be understood as the penalty terms for model updates. In other words, the penalty imposed by BrainUICL on continual individual flow could effectively prevent the model from overfitting to incremental individual target domain $\mathcal{D}_{\mathcal{T}}^i$, while learning new knowledge to improve the model's generalization ability on $\mathcal{D}_G$.

### 2.2 OVERVIEW

Inspired by the rehearsal-based CL methods (Rebuffi et al., 2017; Lopez-Paz & Ranzato, 2017; Castro et al., 2018; Aljundi et al., 2019), which alleviate catastrophic forgetting by replaying a subset of past samples from a storage center, we also adopt a replay-based strategy in this work. As shown in Fig. 2, the workflow of the BrainUICL framework can be divided into three parts when an incremental individual comes in. **First, producing pseudo-labels**: since the incoming subject is without labels, self-supervised learning (SSL) is needed to provide pseudo-labels. We only preserve the confident pseudo-labels whose prediction probabilities are higher than the confidence threshold, for subsequent fine-tuning. **Second, updating incremental models** $\mathcal{M}$: for each batch

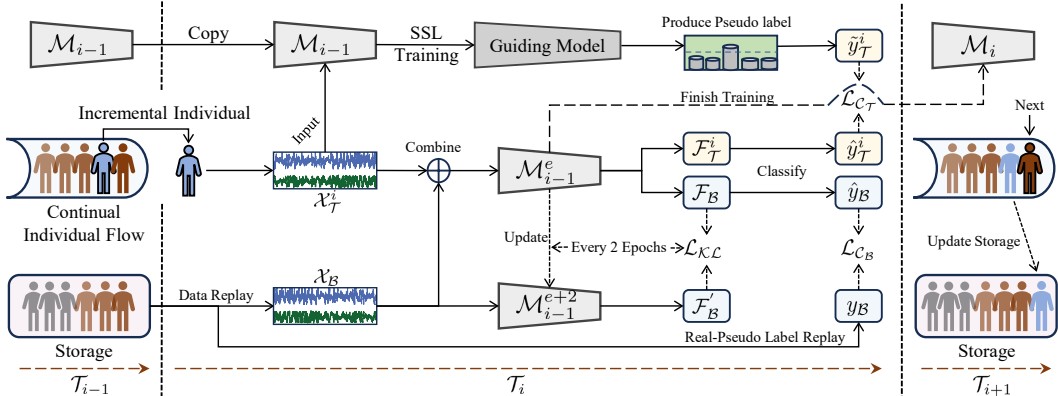

Figure 2: The workflow of the proposed BrainUICL framework. The latest incremental model $\mathcal{M}_{i-1}$ arrives at the $\mathcal{T}_i$ time step. After adapting to the current individual from the continual flow, the model updates from $\mathcal{M}_{i-1}$ to $\mathcal{M}_i$, and is arriving to the next time step $\mathcal{T}_{i+1}$ to adapt to the next individual.

of the incremental individual data $\mathcal{X}_{\mathcal{T}}^i$, the storage center provides a real-pseudo mixed buffer $\mathcal{X}_{\mathcal{B}}$ with the same size for joint training. The details of this procedure are in Sec. 2.3.2. We input $\mathcal{X}_{\mathcal{T}}^i$ and $\mathcal{X}_{\mathcal{B}}$ to the model $\mathcal{M}_{i-1}^e$ and then obtain the hidden features $\mathcal{F}_{\mathcal{T}}^i$ and $\mathcal{F}_{\mathcal{B}}$. Throughing the same classifier, the corresponding prediction $\hat{y}_{\mathcal{T}}^i$ and $\hat{y}_{\mathcal{B}}$ can be obtained, respectively. For the prediction $\hat{y}_{\mathcal{T}}^i$ of the incremental individual, we employ the confident pseudo-label $\tilde{y}_{\mathcal{T}}^i$ generated in the first step to compute the loss $\mathcal{L}_{\mathcal{C}_{\mathcal{T}}}$. Similarly, for buffer's prediction $\hat{y}_{\mathcal{B}}$, we adopt the corresponding replayed label $y_{\mathcal{B}}$ to compute the loss $\mathcal{L}_{\mathcal{C}_{\mathcal{B}}}$. Every two epochs of fine-tuning, we align the hidden features $\mathcal{F}_{\mathcal{B}}$ and $\mathcal{F}_{\mathcal{B}}'$, which are generated from models $\mathcal{M}_{i-1}^e$ and $\mathcal{M}_{i-1}^{e+2}$ at different temporal states, by using the Kullback-Leibler divergence to compute the loss $\mathcal{L}_{\mathcal{KL}}$. Notably, $\mathcal{M}_{i-1}^e$ and $\mathcal{M}_{i-1}^{e+2}$ **denote the incremental model at the e-th and the (e+2)-th fine-tuning epoch**, respectively. More details of this procedure are in Sec. 2.3.3. **Third, updating the storage center**: after adapting to the incremental individual target domain $\mathcal{D}_{\mathcal{T}}^i$, the model has been updated from $\mathcal{M}_{i-1}$ to $\mathcal{M}_i$. Then we utilize the current model $\mathcal{M}_i$ to predict the previous individual's sample $\mathcal{X}_{\mathcal{T}}^i$, and preserve the pseudo-labeled samples with high quality into the storage center.

## 2.3 BrainUICL

In this study, we employ identical model architectures across each downstream EEG task, thereby ensuring equitable validation of the effectiveness of our proposed BrainUICL framework. The model is equipped with a feature extractor to extract EEG features and a temporal encoder to learn the temporal information from the EEG sequences. Given a labeled source domain $\mathcal{D}_S$ (i.e., multiple labeled individual domain, training set), we pretrain the model by minimizing the cross-entropy loss. The detailed model architecture diagram and pretraining process are listed in the Appendix. B. After pretraining the model on the source domain $\mathcal{D}_S$, we have obtained the initial incremental model $\mathcal{M}_0$. Currently, given a continual individual flow (i.e., incremental set), which contains $\mathcal{N}_{\mathcal{T}}$ unlabeled individual target domains, the model $\mathcal{M}_0$ is required to adapt to each individual target domain $\mathcal{X}_{\mathcal{T}}^i$ one by one (i.e., $\mathcal{M}_0 \rightarrow \cdots \rightarrow \mathcal{M}_i \rightarrow \cdots \rightarrow \mathcal{M}_{\mathcal{N}_{\mathcal{T}}}$). After each adaptation, the model is validated on the test domains (i.e., generalization set) to evaluate its generalization ability.

### 2.3.1 SSL TRAINING FOR SUBJECT-SPECIFIC PSEUDO LABEL

Commonly, the existing unsupervised domain incremental learning (Domain-IL) studies (Taufique et al., 2022; Xie et al., 2022; Lamers et al., 2023) employ cluster-based techniques to provide the pseudo-labels in other areas. However, cluster-based are not effective for EEG signals due to their low signal-to-noise ratio (Goldenholz et al., 2009). Considering the sequential nature of EEG signals, we opt for the Contrastive Predictive Coding (CPC) (Oord et al., 2018) algorithm to conduct self-supervised training. Specifically, whenever an incremental individual arrives, we fine-tune the guiding model, which is copied from the latest incremental model $\mathcal{M}_{i-1}$, using its samples with the CPC

algorithm. By doing so, we believe the guiding model can initially fit the incremental individual, thereby producing pseudo labels with higher-quality. Furthermore, we have set a confidence threshold $\xi_1$ to filter out low-quality pseudo-labels. More details about the CPC is listed in the Appendix. C.

### 2.3.2 DYNAMIC CONFIDENT BUFFER

The selection mechanisms of the buffer samples are crucial for those rehearsal-based CL methods. The common option is to store all encountered samples beforehand and randomly select a subset for replay (Castro et al., 2018). Besides, the selection based on FIFO (first-in, first-out) (Taufique et al., 2022), minimum logit distance (Chaudhry et al., 2018a), minimum confidence (Hayes & Kanan, 2021), etc., are also commonly employed for replay. However, these buffer sample selections, which primarily rely on past incremental samples, are not suitable for our UICL setting. Even though we employ the confidence threshold $\xi_1$ to increase the quality of pseudo-labels,

it still inevitably introduces noise, resulting in error accumulation during the fine-tuning stage without the help of true labeled samples. To tackle this, we propose a **selected storage and real-pseudo mixed replay** strategy. Specifically, our storage center consists of two parts: the storage of true labeled samples from the training set $\mathcal{S}_{true} = \{\mathcal{X}_\mathcal{S}, \mathcal{Y}_\mathcal{S}\}$ and the storage of pseudo-labeled samples from the continual individual flow $\mathcal{S}_{pseudo} = \{\mathcal{X}_\mathcal{T}, \tilde{\mathcal{Y}}_\mathcal{T}\}$. At each time step, for the new coming batch of incremental individuals, **we select buffer samples from both $\mathcal{S}_{true}$ and $\mathcal{S}_{pseudo}$ in an 8:2 ratio**, respectively. It can be described as follows:

$$\mathcal{X}_\mathcal{B} = \mathcal{X}_{\mathcal{S} \in \mathcal{S}_{true}} \cup \mathcal{X}_{\mathcal{T} \in \mathcal{S}_{pseudo}} \quad (2)$$

Here, we utilize relatively more real labeled samples from the $S_{true}$, and relatively less previously preserved pseudo-labeled samples from the $S_{pseudo}$ for replay. It can be regarded as another form of penalty terms incorporated on the incremental individuals, as we solely employ a small number of past incremental samples to maintain the diversity of buffer samples. After each round of updating (i.e., $\mathcal{M}_{i-1} \to \mathcal{M}_i$), we utilize the current incremental model $\mathcal{M}_i$ to predict the i-th individual and update its pseudo-labeled samples, whose prediction probability is higher than the confidence threshold $\xi_2$,

---

**Algorithm 1:** UICL Algorithm

**Input:** $\{\mathcal{X}_\mathcal{S}^i, \mathcal{Y}_\mathcal{S}^i\}_{i=1}^{\mathcal{N}_\mathcal{S}}, \{\mathcal{X}_\mathcal{T}^i\}_{i=1}^{\mathcal{N}_\mathcal{T}}, \{\mathcal{X}_G^i, \mathcal{Y}_G^i\}_{i=1}^{\mathcal{N}_G}$
**Output:** $\mathcal{M}$
**Incremental Model Pretraining:**
Pretrain the model $\mathcal{M}_0$ using the source data $\mathcal{X}_\mathcal{S}^i, \mathcal{Y}_\mathcal{S}^i$.
**Unsupervised Individual Continual Learning:**
**for** $i \leftarrow 1$ *to* $\mathcal{N}_\mathcal{T}$ **do**
   Generate the guiding model $\mathcal{M}_g$, copied from
    the latest incremental model $\mathcal{M}_{i-1}$;
   Optimize $\mathcal{M}_g$ by minimizing Eq. (8);
   Generate confident pseudo labels $\tilde{y}_\mathcal{T}^i$ by $\mathcal{M}_g$;
   **if** *i=1* **then**
    |  $\mathcal{X}_\mathcal{B} \leftarrow \mathcal{X}_{\mathcal{S} \in \mathcal{S}_{true}}$;
   **else**
    |  $\mathcal{X}_\mathcal{B} = \mathcal{X}_{\mathcal{S} \in \mathcal{S}_{true}} \cup \mathcal{X}_{\mathcal{T} \in \mathcal{S}_{pseudo}}$;
   **end**
   **for** $j \leftarrow 1$ *to 10* **do**
    Input $\mathcal{X}_\mathcal{B}, \mathcal{X}_\mathcal{T}^i$ to $\mathcal{M}_{i-1}$ and obtain $\hat{y}_\mathcal{B}, \hat{y}_\mathcal{T}^i$;
    Optimize $\mathcal{M}_{i-1}$ by minimizing Eq. (4);
    **if** $j \mid 2 = 0$ **then**
     |  Optimize $\mathcal{M}_{i-1}$ by minimizing Eq. (3);
    **end**
   **end**
   Obtain current incremental model $\mathcal{M}_i$;
   Input $\mathcal{X}_\mathcal{T}^i$ to $\mathcal{M}_i$ and generate confident
    pseudo-labeled samples $(\tilde{\mathcal{X}}_\mathcal{T}^i, \tilde{\mathcal{Y}}_\mathcal{T}^i)$;
   Update storage $\mathcal{S}_{pseudo} = \mathcal{S}_{pseudo} \cup (\tilde{\mathcal{X}}_\mathcal{T}^i, \tilde{\mathcal{Y}}_\mathcal{T}^i)$;
**end**

---

into the storage $\mathcal{S}_{pseudo}$ (i.e., $\mathcal{S}_{pseudo} = \{(\tilde{\mathcal{X}}_\mathcal{T}^0, \tilde{\mathcal{Y}}_\mathcal{T}^0) \cup (\tilde{\mathcal{X}}_\mathcal{T}^1, \tilde{\mathcal{Y}}_\mathcal{T}^1) \cup, ..., \cup (\tilde{\mathcal{X}}_\mathcal{T}^{i-1}, \tilde{\mathcal{Y}}_\mathcal{T}^{i-1}) \cup (\tilde{\mathcal{X}}_\mathcal{T}^i, \tilde{\mathcal{Y}}_\mathcal{T}^i)\}$). Due to the limited number of samples from the source domain and the preservation of only partial samples from incremental individuals, it is acceptable to incur additional storage costs during the continual learning process.

### 2.3.3 CROSS EPOCH ALIGNMENT

During each round of the individual domain adaptation, the incremental model may excessively overfit to some specific individuals without any constraints, which leads to the catastrophic forgetting of previously learned information. This problem can be especially exacerbated if the model encounters outlier individuals whose EEG signals are significantly abnormal. Wang et al. (2022) employed stochastic restoration to randomly restore some tensor elements back to their initial weights. However, this approach may result in certain crucial parameters being completely reset. In our study, we

propose the cross epoch alignment method to overcome the overfitting while taking the preservation of model parameters into consideration. Specifically, given the same incremental model with different temporal states $\mathcal{M}_{i-1}^e$ and $\mathcal{M}_{i-1}^{e+2}$, here $e$ denotes the **current e-th training epoch**. We denote their probability distribution as $\mathcal{P}(\mathcal{M}_{i-1}^e)$ and $\mathcal{P}(\mathcal{M}_{i-1}^{e+2})$, respectively. Every two epochs, we employ Kullback-Leibler (KL) Divergence loss to align the gap between $\mathcal{P}(\mathcal{M}_{i-1}^e)$ and $\mathcal{P}(\mathcal{M}_{i-1}^{e+2})$ as follows:

$$\mathcal{L}_{\mathcal{KL}}(\mathcal{M}_{i-1}, \theta; \mathcal{X}_{\mathcal{B}}) = \mathcal{D}_{\mathcal{KL}}(\mathcal{P}(\mathcal{M}_{i-1}^e) \parallel \mathcal{P}(\mathcal{M}_{i-1}^{e+2})) \tag{3}$$

where $\theta$ denotes the optimization parameters of the model. By aligning the distribution of the previous model state, the network prevents itself from deviating too much even when encountering outlier individuals, enabling the model to achieve better stability. Moreover, avoiding overfitting provides more capacity for further continual domain adaptation, leading to better plasticity.

### 2.3.4 OVERALL LOSS FUNCTION

We use the cross-entropy loss for both buffer samples and incremental individual samples as follows:

$$\mathcal{L}_{\mathcal{C}}(\mathcal{M}_{i-1}, \theta; \mathcal{X}_{\mathcal{B}}, \mathcal{X}_{\mathcal{T}}^i, y_{\mathcal{B}}) = \mathcal{L}_{\mathcal{C}_{\mathcal{B}}} + \alpha\mathcal{L}_{\mathcal{C}_{\mathcal{T}}} = -\sum_c y_{\mathcal{B}_c} \log \hat{y}_{\mathcal{B}_c} + \alpha(-\sum_c \tilde{y}_{\mathcal{T}_c}^i \log \hat{y}_{\mathcal{T}_c}^i) \tag{4}$$

$$\alpha = \begin{cases} 0.01, & i < n \\ 0.1^{(\lg \frac{i}{n})+2}, & i \geq n \end{cases} \tag{5}$$

Here, $\alpha$ is a hyper-parameter that gradually decreases as the continual learning process progresses shown in Fig.3. And $i$ denotes the i-th individual and $n$ represents the number of individuals involved in the training set (i.e., $\mathcal{N}_{\mathcal{S}}$). In other words, the penalty on incremental individuals gradually increases during the CL. Stated differently, this setting is for the model to progressively stabilize itself after it has acquired enough knowledge from the continual individual flow. The overall loss is as follows:

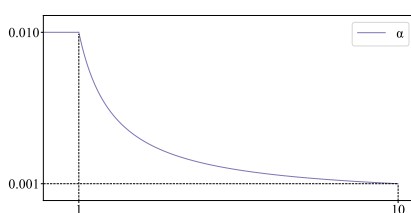

Figure 3: The hyper-parameter $\alpha$ controls the influence of incremental individuals on the model.

$$\mathcal{L}_{overall} = \begin{cases} \mathcal{L}_{\mathcal{C}}, & e \mid 2 \neq 0 \\ \mathcal{L}_{\mathcal{C}} + \mathcal{L}_{\mathcal{KL}}, & e \mid 2 = 0 \end{cases} \tag{6}$$

where $e$ denotes the e-th fine-tuning epoch. The overall algorithm is illustrated in Algorithm. 1.

## 3 EXPERIMENT

### 3.1 EXPERIMENTAL SETUP

As shown in Tab. 1, we employ three mainstream EEG tasks for evaluation: sleep staging, emotion recognition and motor imagery. Specifically, for each EEG task, we conduct our study using a publicly available dataset, namely ISRUC (Khalighi et al., 2016), FACED (Chen et al., 2023), and Physionet-MI (Schalk et al., 2004), respectively.

**ISRUC**, a five-class sleep staging database consists of three sub-groups. We specifically selected sub-group 1, which involves all-night EEG signals from 100 individuals. We excluded subjects 8 and 40 due to some missing channels. The sleep recordings are band-pass filtered (0.3Hz–35Hz) and resampled to 100Hz. **FACED**, an emotion recognition database comprises 32-channel EEG recordings from 123 subjects when they watched 28 emotion-elicitation video clips, and it involves nine emotion categories. The recordings are resampled to 250Hz. **Physionet**, a motor imagery database comprises 64-channel EEG recordings from 109 subjects, covering four motor imagery tasks. We excluded 6 subjects (38, 88, 89, 92, 100, 104) due to differences in the sampling rate or duration of the performed tasks. The recordings are resampled to 160Hz. More details of the datasets are listed in the Appendix. E.

For incremental model pretraining, we set the number of training epoch to 100 and the learning rate is set to 1e-4. For the SSL training and the subsequent fine-tuning, we both set the epoch to 10. The default learning rate for these two process are set to 1e-6 and 1e-7, respectively.

Table 1: Overview of the processed EEG datasets.

| BCI Task | Dataset | Subject | Sampling | Class | Channel | Pretraining | Generalization | Incremental |
|---|---|---|---|---|---|---|---|---|
| Sleep Staging | ISRUC | 98 | 100 | 5 | 8 | 30 | 19 | 49 |
| Emotion Recognition | FACED | 123 | 250 | 9 | 32 | 38 | 24 | 61 |
| Motor Imagery | Physionet | 103 | 160 | 4 | 64 | 32 | 20 | 51 |

Table 2: Overview performance of BrainUICL on three downstream EEG tasks. The results of the Plasticity are evaluated on the incremental set (i.e., continual individual flow) and the results of the Stability are evaluated on the generalization set.

| | Evaluation of Plasticity | | | | | | Evaluation of Stability | | | |
|---|---|---|---|---|---|---|---|---|---|---|
| | Average ACC | | | Average MF1 | | | AAA | | AAF1 | |
| | $\mathcal{M}_0$ | $\mathcal{M}_{i-1}$ | $\mathcal{M}_i$ | $\mathcal{M}_0$ | $\mathcal{M}_{i-1}$ | $\mathcal{M}_i$ | $\mathcal{M}_0$ | $\mathcal{M}_{\mathcal{N}_\mathcal{T}}$ | $\mathcal{M}_0$ | $\mathcal{M}_{\mathcal{N}_\mathcal{T}}$ |
| ISRUC | 65.1 | 72.8 | **75.1 (+10.0)** | 57.6 | 67.1 | **70.0 (+13.4)** | 72.0 | **74.1 (+2.1)** | 69.9 | **72.1 (+2.2)** |
| FACED | 24.2 | 38.9 | **40.3 (+16.1)** | 17.6 | 35.2 | **37.1 (+19.5)** | 24.0 | **36.5 (+12.5)** | 18.7 | **34.5 (+15.8)** |
| Physionet | 46.1 | 47.4 | **48.2 (+2.1)** | 44.6 | 46.3 | **47.4 (+2.8)** | 46.9 | **48.8 (+1.9)** | 46.3 | **48.5 (+2.2)** |

Based on our UICL setting, **each dataset is divided into three parts: pretraining, incremental and generalization sets, with a ratio of 3:5:2.** The pretraining set is used to pretrain the initial incremental model $\mathcal{M}_0$. **The incremental set (i.e., continual individual flow) is used for individual continual domain adaptation and for evaluating the model's plasticity. During this step, the incremental model needs to continuously adapt to each unseen individual one by one.** The generalization set is used to evaluate the model's stability after each round of incremental individual adaptation is completed. The detailed UICL processes are listed in the Appendix. D Fig. 8.

We adopt four metrics to evaluate the stability and the plasticity of our proposed method. For each new incremental individual, we employ **Accuracy (ACC)** and **Macro-F1 (MF1)** to evaluate its performance. Subsequently, we compute the **Average ACC** and **Average MF1** across all incremental individuals involved in the continual individual flow as metrics **to evaluate the plasticity of our model.** After each round of individual domain adaptation, we **evaluate the stability of the updated model on the generalization set using Average Anytime Accuracy (AAA)** (Caccia et al., 2021) and **Average Anytime Macro-F1 (AAF1)** metrics. Here, $AAA_i$ and $AAF1_i$ denote the average ACC and the average MF1 of incremental models $\{\mathcal{M}_0, \mathcal{M}_1, ..., \mathcal{M}_i\}$ on the unseen individuals (i.e., generalization set), respectively. The detailed formulas are as follows:

$$\text{AAA}_i = \frac{1}{i} \sum_{j=1}^{i} \frac{1}{\mathcal{N}_G} \sum_{k=1}^{\mathcal{N}_G} ACC(\hat{\mathcal{Y}}_G^i, \mathcal{Y}_G^i) \qquad \text{AAF1}_i = \frac{1}{i} \sum_{j=1}^{i} \frac{1}{\mathcal{N}_G} \sum_{k=1}^{\mathcal{N}_G} MF1(\hat{\mathcal{Y}}_G^i, \mathcal{Y}_G^i) \qquad (7)$$

where i denotes the i-th incremental individual (i.e., the current individual) and $\mathcal{N}_G$ denotes the number of individual involved in the test domain $\mathcal{D}_G$ (i.e., the generalization set). $\mathcal{Y}_G^i$ and $\hat{\mathcal{Y}}_G^i$ denote the true labels and the corresponding predictions of the model, respectively. Notably, for the subsequent comparison and ablation studies, we conduct multiple runs by randomly shuffling the input order of the continual flow (maintaining the consistency of the data partitions) to conduct a statistical evaluation. Therefore, we calculate the mean and variance of the results(i.e., ACC, MF1, AAA, AAF1) from each run to provide statistical results.

## 3.2 RESULT ANALYSIS

### 3.2.1 OVERVIEW PERFORMANCE

We have conducted our BrainUICL framework on three different downstream EEG tasks shown in Tab. 2. Specifically, **for i-th incremental individual, we compute its personal performance through the same model at three different temporal states (i.e., $\mathcal{M}_0$, $\mathcal{M}_{i-1}$, $\mathcal{M}_i$).** After each adaptation, we measure the latest model's stability on generalization set. Here, $\mathcal{M}_0$ denotes **the initial model**. $\mathcal{M}_{i-1}$ and $\mathcal{M}_i$ represent **the incremental model before and after adapting to the i-th individual**, respectively. $\mathcal{M}_{\mathcal{N}_\mathcal{T}}$ denotes **the final model** after continual adaptation to all incremental individuals. The results demonstrate that our method **can achieve both the better plasticity and stability**. For

plasticity, after each round iteration, the latest model $\mathcal{M}_i$ can improved the performance on the incremental individual compared to the previous state of the model $\mathcal{M}_{i-1}$. When compared to the initial model $\mathcal{M}_0$, there is a significant improvement in the performance of incremental individuals, particularly on the ISRUC and FACED datasets (with 13.4% improvement in average MF1 on ISRUC and 19.5% improvement in average MF1 on FACED). For stability, when most Domain-IL methods simply manage to lower the forgetting rate of prior information, our approach is capable of absorbing new knowledge while further enhancing the model's generalization ability on the generalization set. It can be clearly observed from the comparison of AAA and AAF1 metrics between $\mathcal{M}_0$ and $\mathcal{M}_{\mathcal{N}_\mathcal{T}}$ (e.g., the AAA metric on the FACED dataset between $\mathcal{M}_0$ and $\mathcal{M}_{\mathcal{N}_\mathcal{T}}$: 24.0 vs. 36.5).

### 3.2.2 COMPARISON WITH OTHER METHODS

We have compared our method against several existing **unsupervised domain learning** (UDA), **continual learning** (CL) and **unsupervised continual domain adaptation** (UCDA) methods: **MMD** (Gretton et al., 2006) is a UDA method to match the Maximum Mean Discrepancy distance of feature distributions. **TSTCC** (Eldele et al., 2021) can learn time-series representation from unlabeled data, making it suitable for EEG data. **EWC** (Kirkpatrick et al., 2017) and **LwF** (Li & Hoiem, 2017) are both regularization-based CL methods, applying regularization to prevent the crucial parameters. **UCL-GV** (Taufique et al., 2022) employs FIFO-based buffer and contrastive alignment strategies. **ConDA** (Taufique et al., 2021)adopts a strategy of selectively mixing samples from the incoming batch and buffer data. **CoTTA** (Wang et al., 2022) uses weight-averaged and augmentation averaged prediction and stochastically restore strategies. **ReSNT** (Duan et al., 2023) employ a dynamic memory evolution based replay method to continual decode EEG signals. We implemented these methods based on proposed UICL setting. In practice, the appearance of each new individual in the continual flow is entirely random, and we cannot determine the order in which they arrive. **The difference in the order of continual individual domain adaptation could directly influence the model's learning trajectory**. Therefore, to provide a statistical comparison, we evaluate the stability and robustness of each method under different orders of continual individual flow. Specifically, **we maintained the consistent partitioning of training, incremental, and generalization sets**, and **only altered the input order of the continual individual flow** for each methods by random shuffling, repeated five times in total. Here, **we only report the Plasticity of $\mathcal{M}_i$ state and the Stability of $\mathcal{M}_{\mathcal{N}_\mathcal{T}}$ state**, since each method performs the same in the $\mathcal{M}_0$ state. The statistical results are shown in Tab. 3 and Fig. 4. Compared with other methods, **BrainUICL achieves the best plasticity and stability**. Among the compared methods, UDA-based methods perform the worst. While they could achieve better P on ISRUC and FACED, they dramatically degrade the S. Additionally, on Physionet, both the SP degrades. The performance of CL-based models is slightly better than that of the UDA methods, indicating that continual learning has a greater impact on performance than unsupervised domain adaptation in our UICL setting. In most cases, UCDA-based methods outperform other methods, as they can consider the continuously varying domains. However, UCDA-based methods still fail to achieve better plasticity while simultaneously maintaining the stability. As shown in Fig. 4, the trend of stability changes during each round updating can be visually observed. On the ISRUC and Physionet datasets, all the compared methods exhibit a decline in the AAA and AAF1 curves, except for our method. On the FACED dataset, all the curves demonstrate a fluctuating upward trend; nevertheless, BrainUICL outperforms other methods in the later stages of continual learning. Furthermore, our AAA and AAF1 curves first exhibit a smooth ascending trend and ultimately converge to stability. Moreover, it is worth noting that the confidence intervals of the curves also exhibit a converging trend, with larger intervals at the beginning and ultimately converging to a smaller interval. To sum up, **our BrainUICL demonstrates strong stability and robustness during long-term continual learning, effectively balancing plasticity and stability when compared to other methods.**

### 3.2.3 ABLATION STUDY

To investigate the effectiveness of DCB and CEA modules in BrainUICL, we conducted an ablation study. The ablated methods are as follows: **Base**: both DCB and CEA modules are removed; **Base+CEA**: only DCB module is removed; **Base+DCB**: only CEA module is removed. **BrainUICL**: the framework with all components. Here, we aslo only report the Plasticity of $\mathcal{M}_i$ state and the Stability of $\mathcal{M}_{\mathcal{N}_\mathcal{T}}$ state, since each ablated method performs the same in the $\mathcal{M}_0$ state. The results are shown in Tab. 4 and Fig. 5. Compared with the Base, both the CEA and DCB modules can

Table 3: Performance comparison with existing UDA, CL and UCDA methods.

| | | ISRUC | | | | FACED | | | | Physionet | | | |
|---|---|---|---|---|---|---|---|---|---|---|---|---|---|
| | | ACC | MF1 | AAA | AAF1 | ACC | MF1 | AAA | AAF1 | ACC | MF1 | AAA | AAF1 |
| UDA | MMD | $68.6_{\pm1.8}$ | $62.2_{\pm1.5}$ | $68.1_{\pm0.7}$ | $65.5_{\pm0.9}$ | $34.5_{\pm1.1}$ | $29.7_{\pm1.1}$ | $30.8_{\pm0.7}$ | $27.1_{\pm0.9}$ | $44.5_{\pm0.2}$ | $43.7_{\pm0.2}$ | $45.0_{\pm0.4}$ | $44.4_{\pm0.4}$ |
| | TSTCC | $68.9_{\pm0.8}$ | $63.8_{\pm1.4}$ | $61.3_{\pm1.2}$ | $55.5_{\pm1.7}$ | $37.8_{\pm0.5}$ | $33.7_{\pm0.3}$ | $33.5_{\pm0.5}$ | $30.7_{\pm0.5}$ | $44.9_{\pm1.5}$ | $43.3_{\pm0.2}$ | $45.4_{\pm0.1}$ | $44.1_{\pm0.1}$ |
| CL | EWC | $70.2_{\pm0.6}$ | $65.2_{\pm0.5}$ | $68.4_{\pm0.4}$ | $66.1_{\pm0.5}$ | $37.5_{\pm1.3}$ | $33.3_{\pm1.4}$ | $33.4_{\pm0.7}$ | $30.5_{\pm0.8}$ | $46.9_{\pm0.2}$ | $45.9_{\pm0.1}$ | $46.3_{\pm0.2}$ | $45.4_{\pm0.2}$ |
| | LwF | $71.7_{\pm0.1}$ | $67.0_{\pm0.2}$ | $65.1_{\pm0.2}$ | $59.9_{\pm0.1}$ | $38.3_{\pm0.3}$ | $34.8_{\pm0.4}$ | $34.7_{\pm0.3}$ | $32.3_{\pm0.4}$ | $47.0_{\pm0.3}$ | $45.9_{\pm0.5}$ | $45.8_{\pm0.3}$ | $44.2_{\pm0.6}$ |
| UCDA | UCL-GV | $71.8_{\pm0.3}$ | $66.4_{\pm0.3}$ | $70.7_{\pm0.2}$ | $68.6_{\pm0.2}$ | $38.8_{\pm0.3}$ | $34.8_{\pm0.5}$ | $34.3_{\pm0.3}$ | $31.7_{\pm0.4}$ | $42.7_{\pm0.4}$ | $41.5_{\pm0.3}$ | $42.5_{\pm0.2}$ | $42.0_{\pm0.4}$ |
| | ConDA | $71.6_{\pm0.3}$ | $66.4_{\pm0.3}$ | $70.6_{\pm0.1}$ | $68.5_{\pm0.1}$ | $38.1_{\pm1.2}$ | $34.3_{\pm1.5}$ | $33.9_{\pm0.9}$ | $31.1_{\pm1.1}$ | $45.5_{\pm0.1}$ | $44.4_{\pm0.2}$ | $44.9_{\pm0.2}$ | $43.6_{\pm0.3}$ |
| | CoTTA | $72.2_{\pm0.4}$ | $67.6_{\pm0.3}$ | $69.2_{\pm0.2}$ | $64.7_{\pm0.2}$ | $39.3_{\pm0.6}$ | $35.5_{\pm1.1}$ | $34.7_{\pm0.7}$ | $32.1_{\pm0.9}$ | $47.4_{\pm0.3}$ | $46.3_{\pm0.5}$ | $46.1_{\pm0.3}$ | $44.6_{\pm0.5}$ |
| | ReSNT | $70.7_{\pm0.6}$ | $66.2_{\pm0.7}$ | $71.3_{\pm0.5}$ | $69.4_{\pm0.6}$ | $37.2_{\pm1.3}$ | $33.3_{\pm1.3}$ | $33.8_{\pm0.8}$ | $31.1_{\pm1.1}$ | $45.5_{\pm0.6}$ | $44.5_{\pm0.6}$ | $45.5_{\pm0.1}$ | $44.7_{\pm0.2}$ |
| | BrainUICL | $74.9_{\pm0.2}$ | $69.9_{\pm0.1}$ | $74.0_{\pm0.1}$ | $72.0_{\pm0.1}$ | $40.3_{\pm0.5}$ | $36.8_{\pm0.6}$ | $36.0_{\pm0.5}$ | $33.9_{\pm0.6}$ | $48.4_{\pm0.3}$ | $47.5_{\pm0.3}$ | $48.7_{\pm0.1}$ | $48.3_{\pm0.2}$ |

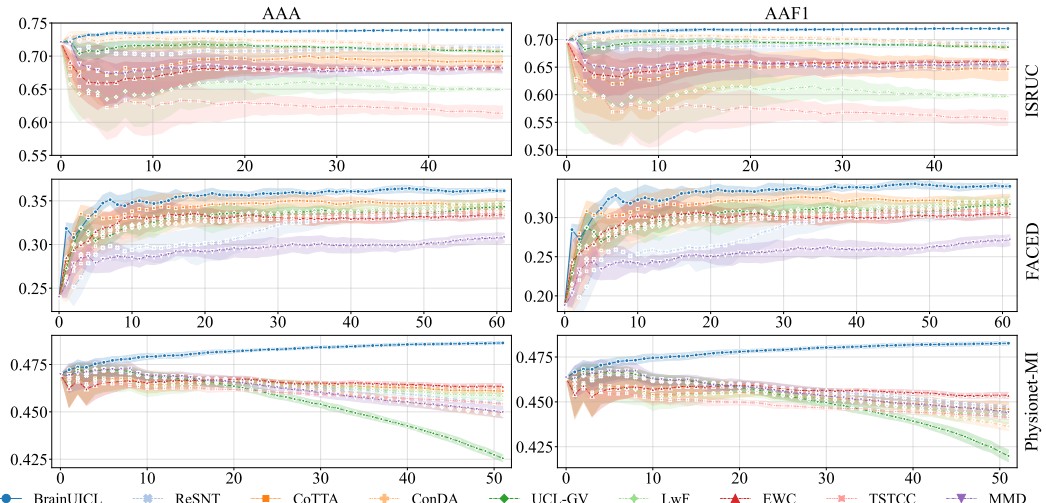

Figure 4: AAA and AAF1 curves of the compared methods and the proposed BrainUICL method. Each point denotes an individual from the continual individual flow, and the middle-line represents the mean value of the AAA and AAF1 metrics under different input orders, while the shaded areas indicate their 95% confidence intervals. Notably, all methods have five same input orders and these orders are randomly different. Our BrainUICL demonstrates the best stability compared to other methods, with a p-value of less than 0.001.

achieve better SP, demonstrating their effectiveness in the UICL setting. For plasticity, DCB module contributes more to our BrainUICL framework compared to CEA in most cases. Only on the average ACC on ISRUC, CEA performs slightly better than DCB (74.2% vs. 73.7%). It is reasonable that the objective of CEA is to prevent the model from overfitting to the newly added individuals, which could result in lower performance on them. **Interestingly, even though we continuously add penalty terms on incremental individuals, the model achieves better plasticity on them**. This can be explained by the fact that if the model has overfitted to some outlier individuals without any constraints, the strong domain shift leads to difficulty for further continual individual domain adaptation. What's worse is that the model may fail to recover and deviate further and further away during the subsequent CL process. For instance, on Physionet, the stability curves of the Base model even surpass those of the CEA and DCB at the beginning of training. However, they consistently decline upon encountering outlier individuals and moreover, fail to recover through subsequent adaptation. For stability, the performances of DCB and CEA are close at the final model state $\mathcal{M}_{\mathcal{N}_T}$, indicating they make roughly equal contributions to the model's stability. Combined with DCB and CEA, BrainUICL outperforms the ablated methods in both plasticity and stability across three datasets. **Furthermore, during long-term continual individual adaptation, our method effectively enables the model to maintain stability even when encountering outliers**. The detailed analysis of the impacts of outliers can be found in the Appendix. H.

Table 4: Performance comparison with ablated methods.

| | ISRUC | | | | FACED | | | | Physionet | | | |
|---|---|---|---|---|---|---|---|---|---|---|---|---|
| | ACC | MF1 | AAA | AAF1 | ACC | MF1 | AAA | AAF1 | ACC | MF1 | AAA | AAF1 |
| Base | $73.3_{\pm0.4}$ | $68.5_{\pm0.4}$ | $73.2_{\pm0.3}$ | $71.2_{\pm0.3}$ | $36.2_{\pm1.1}$ | $31.8_{\pm1.4}$ | $32.6_{\pm0.6}$ | $29.6_{\pm0.9}$ | $47.3_{\pm0.2}$ | $46.5_{\pm0.3}$ | $47.6_{\pm0.3}$ | $47.2_{\pm0.4}$ |
| Base + CEA | $73.9_{\pm0.2}$ | $68.6_{\pm0.1}$ | $73.5_{\pm0.3}$ | $71.6_{\pm0.3}$ | $37.6_{\pm1.3}$ | $33.9_{\pm1.7}$ | $34.3_{\pm0.9}$ | $31.7_{\pm1.2}$ | $47.7_{\pm0.3}$ | $46.8_{\pm0.3}$ | $48.0_{\pm0.1}$ | $47.6_{\pm0.1}$ |
| Base + DCB | $74.1_{\pm0.2}$ | $69.1_{\pm0.3}$ | $73.4_{\pm0.2}$ | $71.4_{\pm0.2}$ | $37.4_{\pm0.8}$ | $33.0_{\pm1.1}$ | $33.4_{\pm0.4}$ | $30.4_{\pm0.4}$ | $48.1_{\pm0.2}$ | $47.4_{\pm0.3}$ | $47.9_{\pm0.3}$ | $47.5_{\pm0.4}$ |
| BrainUICL | $\mathbf{74.9_{\pm0.2}}$ | $\mathbf{69.9_{\pm0.1}}$ | $\mathbf{74.0_{\pm0.1}}$ | $\mathbf{72.0_{\pm0.1}}$ | $\mathbf{40.3_{\pm0.5}}$ | $\mathbf{36.8_{\pm0.6}}$ | $\mathbf{36.0_{\pm0.5}}$ | $\mathbf{33.9_{\pm0.6}}$ | $\mathbf{48.4_{\pm0.3}}$ | $\mathbf{47.5_{\pm0.3}}$ | $\mathbf{48.7_{\pm0.1}}$ | $\mathbf{48.3_{\pm0.2}}$ |

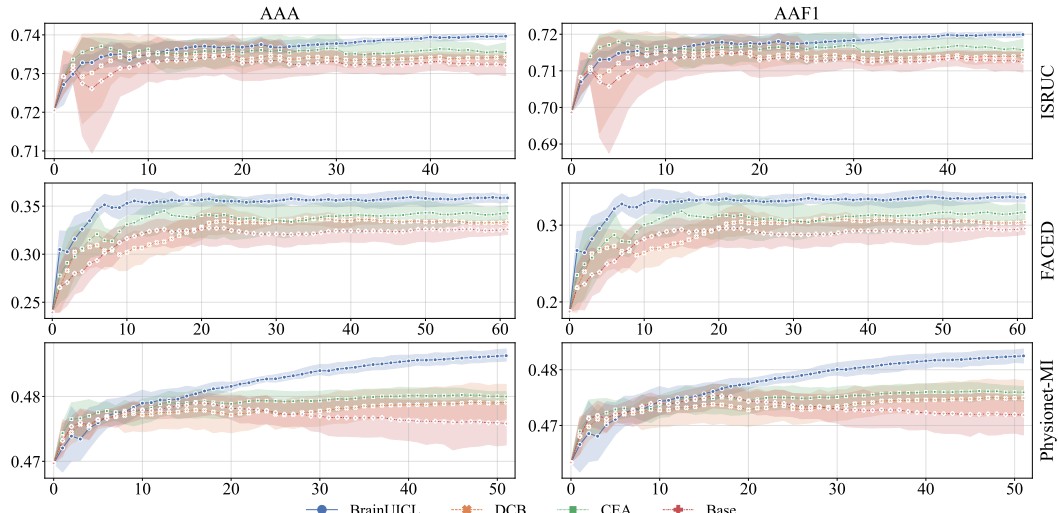

Figure 5: AAA and AAF1 curves of the ablated methods. Each point denotes an individual from the continual individual flow with the middle-line indicating the mean value of the AAA and AAF1 metrics under different input orders, while the shaded areas indicate their 95% confidence intervals. Notably, all methods share five same input orders and these orders are randomly different. The experimental results demonstrate the effectiveness of the proposed DCB and CEA components.

## 4 CONCLUSION

In this work, facing practical applications, we try to make the model not only adapt well to continuously newly incoming subjects, but also generalize well to all the unseen subjects. We propose a novel UICL paradigm for handling EEG tasks in practical applications. And we propose the BrainUICL framework to balance the plasticity-stability dilemma in this setup. The main objective of BrainUICL is to enable the model to continuously adapt well to multiple newly emerging subjects (better P) and simultaneously improve its generalization ability for all unseen subjects (better S), finally becoming a universal expert. We effectively prevent the model from overfitting to incremental individuals during long-term continual individual domain adaptation by increasing the penalty imposed on them. The penalty consists of two parts. First, we employ a selected storage and real-pseudo mixed replay strategy to improve the reliability of replayed EEG samples. Second, we align the incremental model at different temporal states every two epochs to prevent overfitting the model to specific individual distributions. The effectiveness of the proposed BrainUICL has been evaluated on three different downstream EEG tasks. It enables continual individual domain adaptation applications that hold significance in a practical setting.

## 5 ACKNOWLEDGMENTS

This work was supported by STI 2030 Major Projects (2021ZD0200400), the Key Program of the Natural Science Foundation of Zhejiang Province, China (No. LZ24F020004) and the Natural Science Foundation of China (No. 61925603). The corresponding author is Dr. Sha Zhao.

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

## A    RELATED WORK

**EEG Decoding**.  Recently, numerous deep learning-based models have been proposed for EEG tasks(Wu et al., 2013; Pan et al., 2018; Qi et al., 2019; Zhang et al., 2019). For instance, Wang et al. (2024a); Zhou et al. (2024b); Phan et al. (2021); Zhou et al. (2024a) employed EEG-based model for sleep staging, replacing the need for manual scoring. Wang et al. (2024b; 2023); Alturki et al. (2020) utilized EEG-based model to assist in clinical disease diagnosis. Liu et al. (2023b;a) are able to recognize subjects' emotions through EEG signals. However, they overlook the practical situations, as the parameters of these models typically remain fixed after training, leading to limited generalization ability and constraining their application in practice.

**Continual Learning**. Numerous CL methods have been developed to tackle the stability-plasticity (SP) dilemma. Research on continual learning can be categorized into three major streams. **The regularization-based methods:** Kirkpatrick et al. (2017); Zenke et al. (2017); Aljundi et al. (2018); Li & Hoiem (2017); Chaudhry et al. (2018b) directly apply regularization to the parameters to prevent significant changes to those crucial parameters. **The parameter isolation based methods:** Rusu et al. (2016); Mallya & Lazebnik (2018); Fernando et al. (2017) allocate different parameters to different tasks to prevent subsequent tasks from interfering with parameters learned previously. **The rehearsal-based methods:** Rebuffi et al. (2017); Castro et al. (2018); Lopez-Paz & Ranzato (2017); Aljundi et al. (2019) alleviate catastrophic forgetting by replaying a subset of past tasks from a stored memory buffer. Based on these classical CL methods, some of works like Wang et al. (2022); Tang et al. (2021); Saporta et al. (2022) focus on **Continual Domain Adaptation(CDA)** problem, which shares the same setting as ours. UCL-GV Taufique et al. (2022) utilized a contrastive loss to align the gap between the samples in the existing buffer and the gradually varying target domain.

**Continual EEG Decoding**. Recently, existing studies have focused on cross-subject continual EEG decoding. Duan et al. (2023) proposed a dynamic memory evolution based replay method to decode streaming EEG signals. Duan et al. (2024b) proposed a bi-level mutual information maximization based meta optimizer to for sequential EEG classification. Duan et al. (2024a) employed a balanced and informative memory buffer to address this continual EEG decoding challenge.

## B    PRETRAINED MODEL DETAILS

To fairly validate our BrainUICL framework on different downstream EEG tasks, we employ an identical model architecture consisting of three parts: a feature extractor, a feature encoder and a classifier. The feature extractor consists of multiple CNN blocks to extract EEG features, each of which includes a CNN layer, a batch normalization (BN) layer, an activation layer, and a pooling layer(only the first and the fourth CNN layer include the pooling layer). The feature encoder contains multiple TransformerEncoder layers to learn the temporal information from the EEG data. The classifier is composed of several fully connected layers. Notably, we only modified the parameters of the input and output layers to adapt to different EEG tasks. Further details are illustrated in Tab. 8

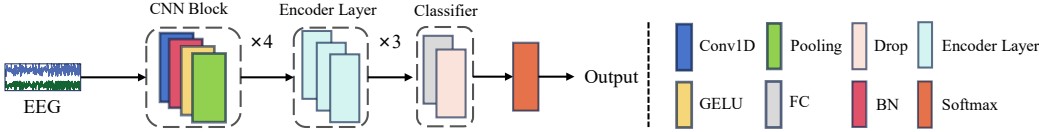

Figure 6: The detailed pre-trained model architecture.

## C    CONTRASTIVE PREDICTIVE CODING

For each arrived individual, we employ a guiding model to produce high-quality pseudo-labels for them, and the pseudo-labels are used for subsequent adaptation. Considering the sequential nature of EEG data, we use the Contrastive Predictive Coding (CPC) algorithm to perform self-supervised fine-tuning on the guiding model, improving the quality of the generated pseudo-labels. Given the latent representation $H = \{h_0, h_1, h_2, h_3, ..., h_t, h_{t+1}, h_{t+2}, h_{t+3}\}$ from the feature encoder, the objective

of CPC is to use the preceding $t$ time steps $H_{i \leq t}$ to predict the subsequent time steps $H_{t \leq i \leq L}$, where $t$ and $L$ denote the predicted time step and the sequence length, respectively. Specifically, we employ a transformer as an autoregressive model to encode $H_{i \leq t}$ into a contextual vector $c_t$. Subsequently, we establish a prediction task in which we utilize linear layers to predict the future EEG time steps, from $h_{t+1}$ until $h_L$, by leveraging the contextual vector $c_t$, such that $z_{t+k} = f_k(c_t)$, where $z_{t+k}^j$ denotes the predicted time steps for $h_{t+k}$. Then, we leverage contrastive loss to update the network. The objective is to specifically align the guiding model with the distribution of the individual target domain. The loss function is as follows:

$$\mathcal{L}_{CPC} = - \underset{H_b}{\mathbb{E}} [log \frac{exp(h_{t+k}^T(f_k(c_t)))}{\sum_{h_j \in H_b} exp((h_j^T f_k(c_t)))}] \tag{8}$$

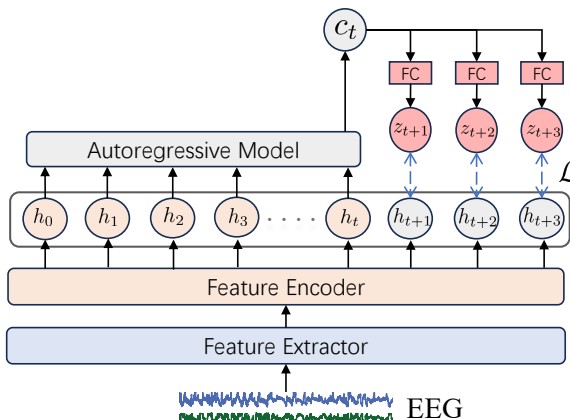

Figure 7: The overview of Contrastive Predictive Coding.

Specifically, when an incremental individual arrives, we first conduct CPC algorithm on the guiding model $M_g$, which is copied from the lastest model $M_{i-1}$, using the incremental individual's samples. After the adaptation, we use the fine-tuned $M_g$ to generate pseudo labels for subsequent training. Specifically, we obtain the classification prediction probabilities(i.e., after the softmax layer) for each sample by inputting the incremental individual samples into guiding model $M_g$. Then, we retain only the high-confidence pseudo-labels with prediction probabilities exceeding the threshold $\xi_1$ for subsequent training. For the threshold $\xi_1$, setting it too high may result in too few generated pseudo-labels, while setting it too low can introduce additional low-quality pseudo-labels. To address this, we conducted a parameter selection experiment to evaluate the impact of different thresholds on the performance of the generated pseudo-labels, ultimately setting $\xi_1$ to 0.9.

## D  UNSUPERVISED INDIVIDUAL CONTINUAL LEARNING SETTING

The detailed process of the UICL is shown in Fig. 8. At the beginning, we initialize the incremental model $\mathcal{M}_0$ using the pretraining set. The incremental model then needs to continuously adapt to each unseen individual one by one. After each round of adaptation, we evaluate the model's stability and plasticity on the generalization set and the latest individual, respectively. For example, the initial model $\mathcal{M}_0$ needs to adapt to the first individual in the continual flow, resulting in the incremental model $\mathcal{M}_1$. We evaluate the stability of the current incremental model $\mathcal{M}_1$ on the generalization set and evaluate the plasticity of the $\mathcal{M}_1$ on the latest individual(i.e., the first individual). After that, the incremental model $\mathcal{M}_1$ needs to adapt to the next individual, and so on.

In sections 4.2.2 and 4.2.3, we assessed the effectiveness of our method under varying input orders of the continual individual flow while maintaining a consistent dataset partition. To facilitate understanding, we provide a simple illustrative example, as shown in the Tab. 5.

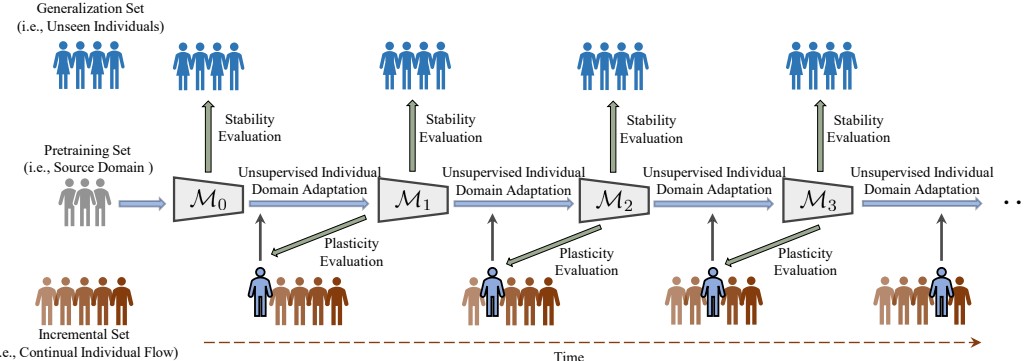

Figure 8: The process of the Unsupervised Individual Continual Learning.

|  | Train Set | Generalization Set | Incremental Set (i.e., Continual Individual Flow) |
|---|---|---|---|
| **Order 1** | 1, 2, 3 | 4, 5 | $6 \rightarrow 7 \rightarrow 8 \rightarrow 9 \rightarrow 10$ |
| **Order 2** | 1, 2, 3 | 4, 5 | $8 \rightarrow 9 \rightarrow 6 \rightarrow 7 \rightarrow 10$ |
| **Order 3** | 1, 2, 3 | 4, 5 | $10 \rightarrow 9 \rightarrow 6 \rightarrow 8 \rightarrow 7$ |
| **Order 4** | 1, 2, 3 | 4, 5 | $9 \rightarrow 8 \rightarrow 6 \rightarrow 7 \rightarrow 10$ |
| **Order 5** | 1, 2, 3 | 4, 5 | $7 \rightarrow 9 \rightarrow 10 \rightarrow 8 \rightarrow 6$ |

Table 5: Overview of Training and Incremental Orders. Here, the numbers denote the different individual IDs.

## E    DATA PREPARATION

**ISRUC:** A sleep dataset consisted of the three sub-groups. We specifically selected sub-group 1, which consists of all-night polysomnography (PSG) recordings from 100 adult individuals and contains 86400 samples. We use six EEG channels (F3-A2, C3-A2, O1-A2, F4-A1, C4-A1, O2-A1) and two EOG channels (E1-M2, E2-M1), and the data is resampled to 100 Hz for evaluation. All EEG signals are divided into 30-second segments, which are then categorized into five distinct sleep stages (Wake, N1, N2, N3, REM) by sleep experts based on the standards set by the American Academy of Sleep Medicine (AASMIber (2007)). The transition patterns between sleep epochs are essential for sleep staging. In line with previous sleep staging studiesPhan & Mikkelsen (2022), we treat this task as a sequence-to-sequence classification problem, defining the sequence length as 20, which corresponds to one sleep sequence consisting of 20 30-seconds samples. We excluded subject 8 and 40 due to some missing channels.

**FACED:** A large finer-grained affective computing EEG dataset covers nine emotion categories (amusement, inspiration, joy, tenderness, anger, fear, disgust, sadness, and neutral emotion) from recordings of 123 subjects. Each recording contains 32-channel EEG signals at 250 Hz sampling rate. All EEG signals are divided into 10-second segments. All the 123 recordings were used for evaluation.

**Physionet-MI:** A motor imagery EEG dataset covers four motor classes (left fist, right fist, both fists and both feet) from recordings of 109 subjects. Each recording contains 64-channel EEG signals at 160 Hz sampling rate. All EEG signals are divided into 4-second segments. All the 109 recordings were used for evaluation.

## F    HYPER-PARAMETER STUDY

### F.1    DYNAMIC CONFIDENT BUFFER

In the Dynamic Confident Buffer, the buffer samples $\mathcal{X}_{\mathcal{B}}$ are selected from both $\mathcal{S}_{true}$ and $\mathcal{S}_{pseudo}$ in an 8:2 radio. In this section, we conduct a hyper-parameter study to validate the effectiveness of our settings shown in Tab. 6.

In most cases, when the selected ratio is set to 8:2, the incremental model can achieve better SP. This suggests that the preference for replaying samples from the true-labeled storage $\mathcal{S}_{true}$ can lead to a better review. The experimental results also indicate that replaying samples from the pseudo-labeled storage $\mathcal{S}_{pseudo}$ in moderation can improve the performances, as it provides more diversity in replaying. In the FACED dataset, the model achieves better plasticity with a ratio of 10:0 than with 8:2. However, it provides much poorer stability without replaying any pseudo-labeled samples from the incremental individuals (35.6% vs. 36.5% in AAA). Compared with the radio of 10:0 and 0:10, the latter performs much worse on both stability and plasticity. This indicates that relying entirely on replaying pseudo-labeled samples can introduce additional noise, leading to error accumulation and forgetting. To sum up, we replay relatively more real samples from the training set to ensure the accuracy of the labels for the replay samples. Meanwhile, we replay a small amount of pseudo-labeled samples produced from the CL process to increase the diversity of the replay samples.

Table 6: Analysis of the $\mathcal{S}_{true}$-$\mathcal{S}_{pseudo}$ Selected Ratio in DCB.

| $\mathcal{S}_{true}$: $\mathcal{S}_{pseudo}$ | ISRUC | | | | FACED | | | | Physionet | | | |
|---|---|---|---|---|---|---|---|---|---|---|---|---|
| | ACC | MF1 | AAA | AAF1 | ACC | MF1 | AAA | AAF1 | ACC | MF1 | AAA | AAF1 |
| 0:10 | 72.6 | 67.1 | 72.6 | 70.6 | 38.5 | 34.3 | 34.3 | 31.7 | 47.0 | 46.1 | 48.0 | 47.4 |
| 2:8 | 72.2 | 66.9 | 72.8 | 70.8 | 39.0 | 35.3 | 35.0 | 32.6 | 47.9 | 47.0 | 48.4 | 47.9 |
| 5:5 | 74.1 | 69.0 | 73.2 | 71.2 | 39.3 | 35.7 | 35.1 | 32.7 | 48.0 | 47.1 | 48.6 | 48.2 |
| 8:2 | **75.1** | **70.0** | **74.1** | **72.1** | 40.3 | 37.1 | **36.5** | **34.5** | **48.2** | **47.4** | **48.8** | **48.5** |
| 10:0 | 74.3 | 69.1 | 73.8 | 71.8 | **40.8** | **37.4** | 35.6 | 33.5 | 48.2 | 47.3 | 48.7 | 48.4 |

## F.2 CROSS EPOCH ALIGNMENT

In the CEA module, The alignment interval can be regarded as a hyper-parameter that controls the impact of the incremental individual on the model. As the alignment interval decreases (e.g., from every two epochs to every epoch), the model performs the alignment operation with the previous model state more frequently. It means the penalty for the impact of incremental individuals is greater and the incremental model is less likely to be affected by new individuals. Meanwhile, as the alignment interval increases (e.g., from every two epochs to every five epochs), the model performs fewer alignment operations, which increases the influence of incremental individuals on the model. To better verify the impact of different selections of the alignment interval, we conducted a hyper-parameter study.

Based on our setting, the training epoch of the fine-tune stage is set to 10. Therefore, we only test the alignment interval from 1(i.e., every epoch) to 5(i.e., every 5 epochs) as the bigger interval is meaningless. The results show that in most cases, when we align with the previous model state every two epochs (i.e., align a total of five times), the incremental model can better balance stability and plasticity. For instance, in FACED, when we perform alignment every 4 epochs, it can achieve better plasticity than every 2 epochs. However, it provides much poorer stability (35.6% vs. 36.5% in AAA and 33.3% vs. 34.5% in AAF1). To sum up, our proposed CEA module aligns the distribution of the previous model states every two epochs. When the model begins to overfit to new individuals, this is mitigated by aligning with the distribution of earlier model states. This approach is beneficial as it effectively prevents the model from overfitting to specific individuals, thereby avoiding a deviation from the original learning trajectory and ensuring the model stability during such long-term continual learning process.

Table 7: Analysis of the Alignment Interval in CEA.

| Alignment Interval | ISRUC | | | | FACED | | | | Physionet | | | |
|---|---|---|---|---|---|---|---|---|---|---|---|---|
| | ACC | MF1 | AAA | AAF1 | ACC | MF1 | AAA | AAF1 | ACC | MF1 | AAA | AAF1 |
| Every Epoch | **75.5** | **70.3** | 73.7 | 71.7 | 39.2 | 35.2 | 35.5 | 33.3 | 47.8 | 47.2 | 48.6 | 48.4 |
| Every 2 Epochs | 75.1 | 70.0 | **74.1** | **72.1** | 40.3 | **37.1** | **36.5** | **34.5** | **48.2** | **47.4** | **48.8** | **48.5** |
| Every 3 Epochs | 74.9 | 70.2 | 73.6 | 71.6 | 40.3 | 36.5 | 35.7 | 33.3 | 48.0 | 47.1 | 48.7 | 48.5 |
| Every 4 Epochs | 74.8 | 70.0 | 73.7 | 71.7 | **40.7** | 36.9 | 35.6 | 33.3 | 48.2 | 47.4 | 48.5 | 48.1 |
| Every 5 Epochs | 74.4 | 69.9 | 73.8 | 71.8 | 39.3 | 35.7 | 35.1 | 32.7 | 48.2 | 47.2 | 48.6 | 48.2 |

The details of the experimental settings for our BrainUICL framework are listed in Tab. 8.

Table 8: Hyper-parameters of the proposed BrainUICL. For Conv1D, the parameters from left to right are: (filter, kernel_size, and stride).

| | | |
|---|---|---|
| Pre-training | Epoch | 100 |
| | Learning Rate | 1e-4 |
| | AdamW $\beta_1$ | 0.5 |
| | AdamW $\beta_2$ | 0.99 |
| | AdamW Weight Decay | 3e-4 |
| | Batch | 32 |
| CNN Blocks | 1-th Conv1D | (64, 50, 6) |
| | 1-th MaxPool1D | (8,8) |
| | 2-th Conv1D | (128, 8) |
| | 3-th Conv1D | (256, 8) |
| | 4-th Conv1D | (512, 8) |
| | 4-th MaxPool1D | (4, 4) |
| Transformer | Attention Head | 8 |
| | Attention Dim | 512 |
| | Attention Layer | 3 |
| | Dropout | 0.1 |
| Self-supervised Learning | Epoch | 10 |
| | Learning Rate | 1e-6 |
| Continual Adaptation | Epoch | 10 |
| | Learning Rate | 1e-7 |
| | Confident Threshold $\xi_1$ | 0.9 |
| | Confident Threshold $\xi_2$ | 0.9 |
| | Alignment Interval | 2 |

## G  COMPUTATIONAL COST

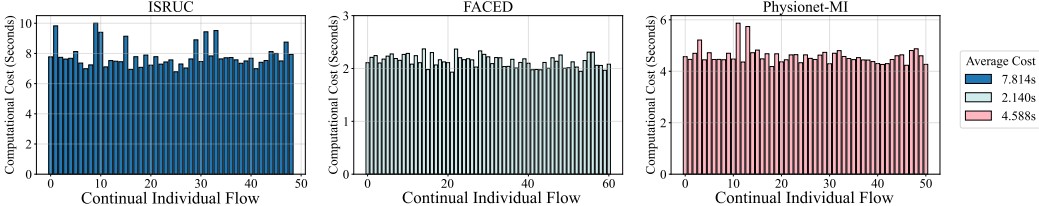

Figure 9: The computational cost per individual.

To assess the computational efficiency of our proposed BrainUICL framework, we conducted a comprehensive analysis of the time cost per individual across three diverse datasets, as illustrated in Fig. 9. Our BrainUICL framework enables the model to rapidly adapt to an unseen individual, with an average processing time of just a few seconds. This rapid adaptation capability is a crucial feature of our BrainUICL framework, positioning it as an ideal solution for real-world applications that demand quick and seamless integration.

## H  IMPACT OF OUTLIERS

We have listed the performance changes of partial outliers and their impact on the incremental model on the ISRUC dataset in Tab. 9. As we can see, the initial performance of the model $\mathcal{M}_0$ on these outliers is quite low. Then, with the model continuously absorbs the new knowledge, before the adaptation $\mathcal{M}_{i-1}$, the performance of these outliers has already seen a significant improvement compared to the initial state $\mathcal{M}_0$. After adaptation, their performance is further enhanced $\mathcal{M}_i$. Meanwhile, the model's generalization ability is also steadily increasing, demonstrating that our method can not only improve the performance on outliers, but also achieve better stability after each adaptation.

## I  PERFORMANCE VARIATIONS IN TRAIN SET

In this section, we evaluate the performance variations of the training set throughout the continual learning process, as illustrated in Fig. 10. The training set is used solely for pretraining the initial

Table 9: The performance changes of partial outliers and their impact on the model on the ISRUC dataset. Here, ID denotes the position of outliers in the continuous individual flow. $\mathcal{M}_0$ denotes the initial model. $\mathcal{M}_{i-1}$ and $\mathcal{M}_i$ denote the incremental model before and after adapting to the current individual, respectively.

| | Evaluation of Plasticity | | | | | | Evaluation of Stability | | | |
| | Individual ACC | | | Individual MF1 | | | AAA | | AAF1 | |
| Outlier ID | $\mathcal{M}_0$ | $\mathcal{M}_{i-1}$ | $\mathcal{M}_i$ | $\mathcal{M}_0$ | $\mathcal{M}_{i-1}$ | $\mathcal{M}_i$ | $\mathcal{M}_{i-1}$ | $\mathcal{M}_i$ | $\mathcal{M}_{i-1}$ | $\mathcal{M}_i$ |
|---|---|---|---|---|---|---|---|---|---|---|
| ID = 6 | 35.93 | 46.51 | **53.37** | 19.62 | 36.01 | **48.96** | 72.87 | **73.08** | 70.91 | **71.12** |
| ID = 10 | 28.87 | 45.75 | **60.85** | 9.92 | 38.59 | **52.78** | 73.23 | **73.29** | 71.24 | **71.32** |
| ID = 12 | 36.07 | 41.90 | **46.31** | 10.60 | 37.93 | **39.97** | 73.37 | **73.47** | 71.39 | **71.53** |
| ID = 24 | 34.06 | 49.38 | **50.42** | 24.04 | 40.97 | **41.47** | 73.47 | **73.51** | 71.48 | **71.53** |
| ID = 25 | 37.86 | 52.74 | **63.33** | 10.98 | 46.53 | **52.90** | 73.51 | **73.56** | 71.53 | **71.59** |
| ID = 27 | 24.87 | 39.49 | **54.62** | 9.58 | 34.51 | **49.99** | 73.60 | **73.62** | 71.63 | **71.64** |
| ID = 37 | 20.83 | 55.42 | **56.25** | 14.52 | 47.97 | **51.81** | 73.88 | **73.90** | 71.90 | **71.91** |
| ID = 40 | 28.54 | 66.45 | **76.56** | 20.37 | 62.01 | **69.86** | 73.99 | **74.03** | 72.00 | **72.05** |

incremental model $M_0$ and does not participate in the subsequent continual learning process. We do not analyze the results on the Physionet-MI dataset, as the initial model $M_0$ has already demonstrated high performance on this dataset. In contrast, on the ISRUC and FACE datasets, the model's performance on the training set shows an overall improvement, rather than the catastrophic forgetting typically associated with continual learning. This is reasonable, given that 80% of the samples we replay are sourced from the training set, which enhances performance as we continuously replay the labeled samples from the training set.

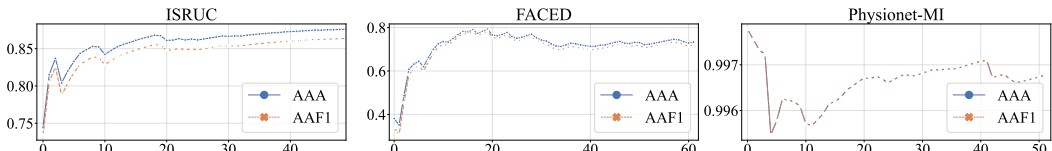

Figure 10: The performance variations of the train set during the continual learning.

## J COMPARED WITH OTHER MEMORY SAMPLING METHODS

To validate the effectiveness of the proposed DCB-based memory sampling approach, we conduct a comparative study with other popular memory sampling methods: FIFO (i.e., First-In-First-Out), RS (i.e., reservoir sampling), and Uniform (i.e., uniform random sampling). Notably, in this study, we only replace our DCB-based memory sampling method with the other methods, maintaining the consistency of other components to ensure a fair comparison. Our method significantly outperforms the compared methods, as illustrated in Tab. 10 and Fig. 11, particularly on the FACE and Physionet-MI datasets. Overall, the FIFO-based approach performs the worst, as it relies heavily on data from the previous individual for replay. When encountering outliers, the model's performance inevitably declines and may not be recoverable (see the FIFO curve in Physionet). On the Physionet dataset, the UCLGV method using the FIFO setting exhibits a similar downward trend, as shown in Fig. 4. Among the comparison methods, the uniform approach performs the best. This is because, although we save all newly added individual samples into storage, the number of true labeled samples from the training set remains significantly higher than that of pseudo-labeled samples during the early stages of training. Consequently, the randomly sampled replay samples are predominantly accurately labeled. However, in the later stages of training, as pseudo-labeled samples are continuously added to storage without filtering, each replay introduces a substantial number of low-quality samples, resulting in a decline in model performance (see the Uniform curve in ISRUC).

Table 10: Performance comparison with other memory sampling methods is presented. Notably, FIFO refers to First-In-First-Out, RS denotes reservoir sampling, and Uniform indicates uniform random sampling, respectively.

|  | ISRUC | | | | FACED | | | | Physionet-MI | | | |
|---|---|---|---|---|---|---|---|---|---|---|---|---|
|  | ACC | MF1 | AAA | AAF1 | ACC | MF1 | AAA | AAF1 | ACC | MF1 | AAA | AAF1 |
| FIFO | 70.5 | 65.6 | 71.3 | 69.0 | 34.9 | 29.6 | 30.4 | 26.8 | 43.1 | 41.9 | 43.9 | 43.2 |
| RS | 71.2 | 65.8 | 70.7 | 68.6 | 33.4 | 28.8 | 30.7 | 27.0 | 44.8 | 43.4 | 45.7 | 44.7 |
| Uniform | 74.2 | 68.7 | 73.4 | 71.4 | 37.8 | 33.3 | 33.1 | 30.5 | 47.3 | 46.3 | 47.7 | 47.5 |
| Ours (DCB) | **75.1** | **70.0** | **74.1** | **72.1** | **40.3** | **37.1** | **36.5** | **34.5** | **48.2** | **47.4** | **48.8** | **48.5** |

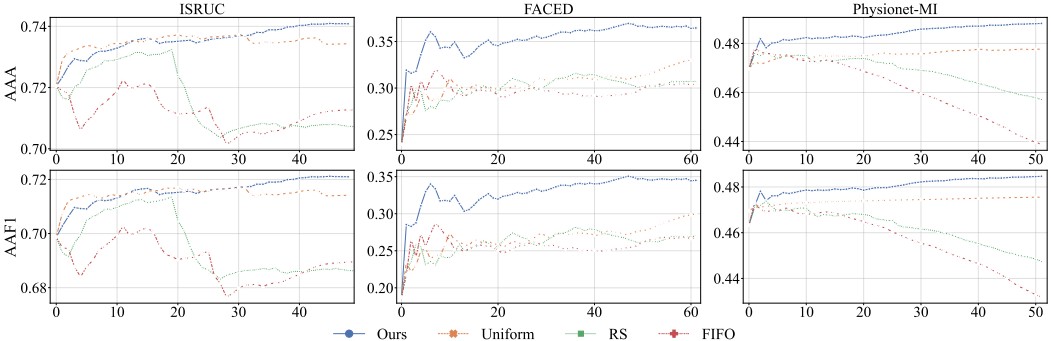

Figure 11: AAA and AAF1 curves of the compared memory sampling methods and our DCB method.

# K PARTITION STUDY

In sections 4.2.2 and 4.2.3, we assessed the effectiveness of our method under varying input orders of the continual individual flow while maintaining a consistent dataset partition. To evaluate the effectiveness of our proposed method across different dataset partitions, we conducted a partition study. In this study, while keeping other experimental settings unchanged, we randomly shuffled the dataset partitions (i.e., pretraining set, incremental set, generalization set) for experimentation, repeating the process three times, as shown in Tab. 11 and Fig. 12. The experimental results demonstrate that our method consistently exhibits strong performance across various dataset partitions, remaining unaffected by the specific partitioning of the dataset.

Table 11: Overview performance of BrainUICL on three EEG tasks under different partition.

|  |  | Evaluation of Plasticity | | | | | | Evaluation of Stability | | | |
|---|---|---|---|---|---|---|---|---|---|---|---|
|  |  | Average ACC | | | Average MF1 | | | AAA | | AAF1 | |
|  |  | $\mathcal{M}_0$ | $\mathcal{M}_{i-1}$ | $\mathcal{M}_i$ | $\mathcal{M}_0$ | $\mathcal{M}_{i-1}$ | $\mathcal{M}_i$ | $\mathcal{M}_0$ | $\mathcal{M}_{\mathcal{N}_\mathcal{T}}$ | $\mathcal{M}_0$ | $\mathcal{M}_{\mathcal{N}_\mathcal{T}}$ |
| ISRUC | Partition1 | 67.5 | 72.6 | **74.4 (+6.9)** | 60.0 | 67.9 | **70.4 (+10.4)** | 68.9 | **73.4 (+4.5)** | 65.6 | **70.9 (+5.3)** |
|  | Partition2 | 65.3 | 72.9 | **74.5 (+9.2)** | 57.8 | 67.1 | **69.6 (+11.8)** | 71.9 | **74.7 (+2.8)** | 69.0 | **72.0 (+3.0)** |
|  | Partition3 | 65.0 | 72.1 | **73.6 (+8.6)** | 56.3 | 66.9 | **69.0 (+12.7)** | 72.5 | **76.6 (+4.1)** | 70.3 | **74.8 (+4.5)** |
|  | Original | 65.1 | 72.8 | **75.1 (+10.0)** | 57.6 | 67.1 | **70.0 (+13.4)** | 72.0 | **74.1 (+2.1)** | 69.9 | **72.1 (+2.2)** |
| FACED | Partition1 | 23.6 | 36.9 | **37.1 (+13.5)** | 16.9 | 35.6 | **33.1 (+16.2)** | 25.4 | **35.7 (+10.3)** | 20.8 | **33.2 (+12.4)** |
|  | Partition2 | 23.8 | 38.6 | **39.2 (+15.4)** | 17.3 | 34.6 | **35.1 (+17.8)** | 24.9 | **37.1 (+12.2)** | 19.8 | **34.6 (+14.8)** |
|  | Partition3 | 24.1 | 38.8 | **39.3 (+15.2)** | 17.5 | 35.5 | **35.7 (+18.2)** | 24.1 | **35.6 (+11.5)** | 18.3 | **33.2 (+14.9)** |
|  | Original | 24.2 | 38.9 | **40.3 (+16.1)** | 17.6 | 35.2 | **37.1 (+19.5)** | 24.0 | **36.5 (+12.5)** | 18.7 | **34.5 (+15.8)** |
| Physionet-MI | Partition1 | 44.5 | 45.6 | **45.9 (+1.4)** | 43.0 | 44.5 | **44.9 (+1.9)** | 50.9 | **52.5 (+1.6)** | 50.4 | **52.3 (+1.9)** |
|  | Partition2 | 47.4 | 49.8 | **50.1 (+2.7)** | 46.0 | 48.7 | **49.4 (+3.4)** | 43.3 | **44.2 (+0.9)** | 42.9 | **44.0 (+1.1)** |
|  | Partition3 | 45.6 | 46.7 | **48.2 (+2.6)** | 44.4 | 45.6 | **47.4 (+3.0)** | 48.1 | **49.9 (+1.8)** | 47.6 | **49.6 (+2.0)** |
|  | Original | 46.1 | 47.4 | **48.2 (+2.1)** | 44.7 | 46.3 | **47.4 (+2.7)** | 46.9 | **48.8 (+1.9)** | 46.3 | **48.5 (+2.2)** |

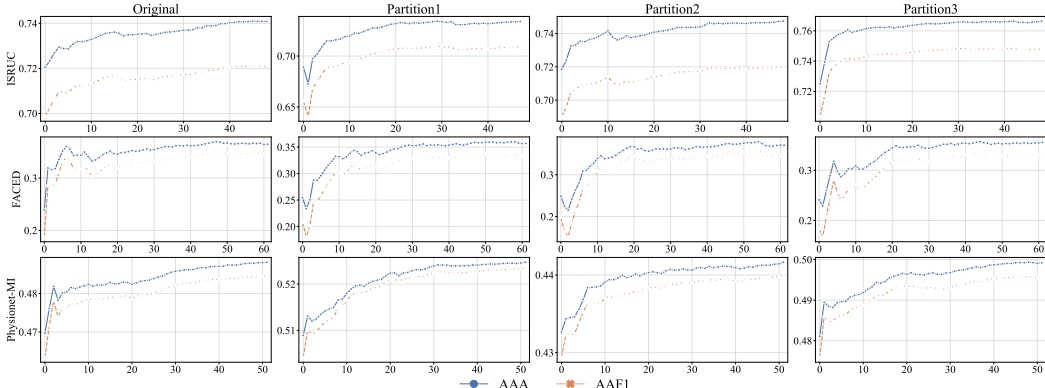

Figure 12: AAA and AAF1 curves of the our methods with different dataset partitions.

## L    FUTURE WORK

In current practice, many EEG-related traditional manual assessments have been replaced by deep learning based models. These models are typically trained on a source domain and then applied to practical testing. However, there are many limitations within this application. On the one hand, the model's generalization performance is limited due to constraints on the size of the source domain. On the other hand, the model with fixed parameters may not adapt to each unseen individual due to individual discrepancies. To address this issue, we proposed the BrainUICL framework which enables the EEG-based model to continuously adapt to newly appearing subjects, while simultaneously strengthening its generalization ability for those unseen subjects. On the downside, we have only applied our proposed BrainUICL framework to three mainstream EEG tasks (i.e. sleep staging, emotion recognition, and motor imagery). The primary reason is the limited size of publicly available datasets for other EEG tasks, which typically only include a few dozen individuals at most. In future work, in addition to the aforementioned three EEG tasks, we intend to extend our proposed BrainUICL framework to include a broader range of practical EEG-based tasks (e.g., Major Depressive Disorder Diagnosis, Fatigue Detection, Disorders of Consciousness Diagnosis).

