# OpenReview forum: "BrainUICL: An Unsupervised Individual Continual Learning Framework for EEG Applications"
_ICLR.cc/2025/Conference — ICLR 2025 Poster_

### Official Review · Reviewer_J7FP · 2024-11-03

**Soundness:** 2
**Presentation:** 3
**Contribution:** 2
**Rating:** 5
**Confidence:** 5

**Summary:**

Pre-trained EEG models often cannot be effectively generalized in practice due to high inter-subject variability. In this work, a novel unsupervised continual learning (CL) approach is proposed that aims to balance adaptation and generalization. To mitigate catastrophic forgetting, the method introduces a penalty term based on cross-epoch alignment and uses a dynamic confident buffer to preserve prior knowledge. Experiments conducted on three different datasets demonstrate superior performance.

**Strengths:**

The research question addressed in the paper is interesting. The results show that the proposed approach outperforms existing methods.

**Weaknesses:**

1.	The selection mechanism for the buffer samples requires further clarification, particularly with regard to the number of samples retained per individual. To strengthen the evaluation, it would be helpful to compare the effectiveness of the proposed approach with standard memory sampling techniques, such as reservoir sampling, as well as recent advanced methods specifically designed to address inter-subject variability in EEG data

2.	The KL-based penalty term needs further clarification, in particular why it is only applied in every second epoch and not in every training epoch. Furthermore, the mechanism that controls the impact of this penalty term remains unclear. Is there a specific parameter that controls this loss term to regulate its influence during training?

3.	How the datasets are divided into source, target and test sets is unclear. Given the heterogeneity caused by inter-subject variability, if subjects were randomly assigned to each set (source, target, test), conducting the experiments in multiple runs and reporting the averaged accuracy would be advantageous.

**Questions:**

1.	Clarification is needed on how the threshold for self-supervised learning (SSL) is determined in the presence of inter-subject data heterogeneity. How effective are the generated pseudo-labels given this variability? Are there specific criteria for setting this threshold? Additionally, considering that the previous model may be biased toward earlier subjects, could inter-subject variability lead to inaccuracies in the pseudo-labels?
2.	How is the plasticity of the incremental set evaluated? Is there a specific incremental split for training and testing?
3.	What is the total number of samples stored in the storage buffer for each individual? In addition, how are the samples of the target domain replaced in the memory?

---

> ### Author Response · Authors · 2024-11-20
> **Response to Reviewer J7FP[1/N]**
>
> **Many thanks for your valuable and insightful suggestions**. In this rebuttal, we aim to address each of the key issues and points you have raised.
>
> ---
>
> **Q1:** The selection mechanism for the buffer samples requires further clarification, particularly with regard to the number of samples retained per individual.
>
> **R1:** Thanks for your valuable concerns. We will address your concerns from the following two perspectives:
>
> 1. **Buffer Sample Selection:** In our DCB module, we design two distinct storage: $S_{true}$ and $S_{pseudo}$. Here, $S_{true}$={$X_S,Y_S$} refers to the storage of true labeled samples from the training set, while $S_{pesudo}$={$X_T,\tilde{Y_T}$} denotes the pseudo-labeled samples generated during the CL process. We utilize a greater proportion of real labeled samples from $S_{true}$​ and a smaller proportion of previously preserved pseudo-labeled samples from $S_{pseudo}$​ for replay, specifically in an 8:2 ratio, as determined through a hyperparameter study detailed in Appendix E1.1, Tab 5 (**page 17**). This approach allows us to select more true labeled samples from $S_{true}$​ to ensure the accuracy of replay. Simultaneously, we replay a limited number of pseudo-labeled samples from $S_{pseudo}$​ to enhance the diversity of the replay samples.
>
> 2. **Individual Sample Retention:** After the incremental model has adapted to a new individual, we only save the high-quality samples—those with a prediction probability exceeding the high-confidence threshold ξ2​ (0.9)—into the storage $S_{pseudo}$​ for subsequent replay. By setting such a confidence threshold to filter out low-quality samples, the number of samples retained for each incremental individual in $S_{pseudo}$​ remains uncertain. For individuals to which the model adapts well, a larger number of high-confidence pseudo-labeled samples are saved. In contrast, for individuals that the model struggles to fit, fewer pseudo-labeled samples are retained.
>
>
> We hope these clarifications will address your concerns. Thank you once again for your valuable feedback.

---

> ### Author Response · Authors · 2024-11-20
> **Response to Reviewer J7FP[2/N]**
>
> **Q2:** it would be helpful to compare the effectiveness of the proposed approach with standard memory sampling techniques, such as reservoir sampling, as well as recent advanced methods specifically designed to address inter-subject variability in EEG data.
>
> **R2:** Thanks for your insightful suggestions. We'd like to address your concerns from the following two perspectives:
>
> 1. **Compared with other Memory Sampling Techniques:** We have added a new comparative study with other popular memory sampling methods (e.g., FIFO, Reservoir Sampling, Uniform Random Sampling). The comparative results are illustrated in the table below:
>
>   | Dataset | Method | ACC | MF1 | AAA | AAF1 |
>   | --- | --- | --- | --- | --- | --- |
>   | ISRUC | FIFO | 70.5 | 65.6 | 74.1 | 72.1 |
>   |     | RS  | 71.2 | 65.8 | 70.7 | 68.6 |
>   |     | Uniform | 74.2 | 68.7 | 73.4 | 71.4 |
>   |     | **Ours (DCB)** | **75.1** | **70.0** | **74.1** | **72.1** |
>   | FACED | FIFO | 34.9 | 29.6 | 30.4 | 26.8 |
>   |     | RS  | 33.4 | 28.8 | 30.7 | 27.0 |
>   |     | Uniform | 37.8 | 33.3 | 33.1 | 30.5 |
>   |     | **Ours (DCB)** | **40.3** | **37.1** | **36.5** | **34.5** |
>   | Physionet-MI | FIFO | 43.1 | 41.9 | 43.9 | 43.2 |
>   |     | RS  | 44.8 | 43.4 | 45.7 | 44.7 |
>   |     | Uniform | 47.3 | 46.3 | 47.7 | 47.5 |
>   |     | **Ours (DCB)** | **48.2** | **47.4** | **48.8** | **48.5** |
>
>   The results demonstrate that our method significantly outperforms the compared approaches, thereby validating the effectiveness of our proposed selective replay strategy. Specifically, these memory sampling methods are not well-suited for long-term individual continual learning, as they can easily introduce outlier samples, causing the incremental model to deviate excessively from its original learning trajectory. Consequently, the proposed DCB method addresses the requirements for replay samples in long-term individual continual learning, **ensuring both high quality and diversity among the replay samples.** For a more detailed analysis and the AAA/AAF1 variation curves, please refer to the newly uploaded file, Appendix I, Tab. 9, and Fig. 11 (**page 20**).
>
> 2. **Compared with Recent Continual EEG Decoding Method:** We have included a recent cross-subject EEG-based continual learning method, ReSNT[1], for comparison. Since ReSNT is a supervised continual learning method, we made modifications during the reproduction process to enable it to function within our proposed unsupervised individual continual learning framework. Specifically, when an incremental individual arrives, we apply our SSL method (i.e., CPC) to generate high-confidence pseudo-labels for subsequent supervised fine-tuning of ReSNT. A statistical evaluation of ReSNT across all the datasets is presented in Tab. 3 (**page 9**). Our model significantly outperforms ReSNT on all the datasets.
>
>
> We hope these additional comparative studies will address your concerns. Thank you once again for your valuable suggestions.
>
> [1] Replay with Stochastic Neural Transformation for Online Continual EEG Classification[C]//2023 IEEE International Conference on Bioinformatics and Biomedicine (BIBM). IEEE, 2023: 1874-1879.

---

> ### Author Response · Authors · 2024-11-20
> **Response to Reviewer J7FP[3/N]**
>
> **Q3:** The KL-based penalty term needs further clarification, in particular why it is only applied in every second epoch and not in every training epoch. Furthermore, the mechanism that controls the impact of this penalty term remains unclear. Is there a specific parameter that controls this loss term to regulate its influence during training?
>
> **R3:** Many thanks for your insightful questions. We'd like to address your concerns from the following perspectives:
>
> 1. **Further Clarification for KL-based Penalty:** The core idea of BrainUICL is to impose a penalty on incremental individuals to prevent the model from overfitting to them and forgetting previously acquired knowledge. Accordingly, we propose the Cross Epoch Alignment (CEA) module to implement a soft penalty (i.e., KL-based penalty) on incremental individuals. Specifically, we align the distribution of the previous model states every two epochs. When the model begins to overfit to new individuals, this is mitigated by aligning with the distribution of earlier model states. This approach is beneficial as it effectively prevents the model from overfitting to specific individuals(especially outliers, this part of analyse is listed in Appendix. G), thereby avoiding a deviation from the original learning trajectory and ensuring the model stability during such long-term continual learning process.
> 2. **The Impact of the Alignment Interval:** In the CEA module, the alignment interval can be regarded as a hyper-parameter to control the impact of this penalty. As the alignment interval decreases (e.g., from every two epochs to every epoch), the model performs the alignment operation with the previous model state more frequently. It means the penalty for the incremental individuals is greater and the incremental model is less likely to be affected by new individuals. Meanwhile, as the alignment interval increases (e.g., from every two epochs to every five epochs), the model performs fewer alignment operations, which increases the influence of incremental individuals on the model.
> 3. **The Selection of the Alignment Interval:** Furthermore, we conducted a hyperparameter study to assess the impact of different selections for the alignment interval (see Appendix. E.2 **page 17**). The results indicate that the performance is optimal when the alignment is operated every two epochs.
>
> We hope these clarifications will address the reviewer's concerns. Thanks for your insightful feedbacks.
>
> ---
>
> **Q4:** How the datasets are divided into source, target and test sets is unclear.
>
> **R4:** Thanks for your concern. In our UICL setting, each dataset is randomly divided into three parts: pretraining(i.e., source), incremental(i.e., target) and generalization(i.e., test) sets, with a ratio of 3: 5: 2. The number of participants in each specific set is displayed in Tab.1 (**page 7**) and the detailed explanations are listed in Section. 4.1, Experimental Setup (**page 7**).
>
> ---
>
> **Q5:** Given the heterogeneity caused by inter-subject variability, if subjects were randomly assigned to each set (source, target, test), conducting the experiments in multiple runs and reporting the averaged accuracy would be advantageous.
>
> **R5:** Thanks for your constructive comment. We have added a partition study to evaluate the effectiveness of our proposed method across different datasets. While maintaining other experimental settings unchanged, we randomly shuffled the dataset partitions (i.e., pretraining set, incremental set, generalization set) for experimentation, repeating the process three times. We provide the model's performance on three datasets under different data partitions. More details can be found in the Appendix. J, Tab. 10, and Fig. 12 (**page 21**), for detailed experimental results. The results indicate that our model consistently achieves improved stability and plasticity across various initial dataset partitions, confirming that its performance is not influenced by the initial data partitioning.
>
> In this study, we do not report the average performance across different runs, as this would lack statistical significance due to variations in the initial model $M_0$​ performance (which is pretrained on different source data), differences in the individuals within the incremental set, variations in the input order of the continual flow, and the distinct generalization sets utilized to assess stability.
>
> We hope this additional partition study will address your concern. Thank you again for your valuable comment.

---

> ### Author Response · Authors · 2024-11-20
> **Response to Reviewer J7FP[4/N]**
>
> **Q6:** Clarification is needed on how the threshold for self-supervised learning (SSL) is determined in the presence of inter-subject data heterogeneity. How effective are the generated pseudo-labels given this variability? Are there specific criteria for setting this threshold?
>
> **R6:** Many thanks for your valuable concern. We have included a detailed description of the SSL mechanism in Appendix B (**page 15**), which covers the process of generating pseudo label confidence values, the generation of pseudo labels, and the criteria for selecting the confidence threshold. The details are as follows:
>
> 1. **Generating Pseudo Labels:** When an incremental individual arrives, we first apply the CPC algorithm to the guiding model $M_g$​, which is a copy of the most recent model $M_{i−1}$​, using the samples from the incremental individual. After adaptation, we utilize the fine-tuned guiding model​ to generate pseudo labels for subsequent training. Specifically, we obtain classification prediction probabilities (i.e., confidence values) for each sample by inputting the incremental individual samples into the guiding model $M_g$​ after the softmax layer. We then retain only those high-confidence pseudo labels with prediction probabilities exceeding the threshold $\xi_1$​ (0.9) for further training.
>
> 2. **Selecting the Confidence Threshold:** For the threshold $\xi_1$​, setting it too high may result in an insufficient number of generated pseudo labels, while setting it too low can introduce additional low-quality pseudo labels. To address this issue, we conducted a parameter selection experiment to evaluate the impact of different thresholds (0.75, 0.80, 0.85, 0.90, 0.95) on the performance of the generated pseudo labels. The experimental results indicate that the optimal performance is achieved when the confidence threshold $\xi_1$ is set to 0.90.
>
>
> We hope these additional clarifications will address your concerns.
>
> ---
>
> **Q7: Additionally, considering that the previous model may be biased toward earlier subjects, could inter-subject variability lead to inaccuracies in the pseudo-labels?**
>
> **R7:** Thank you for your insightful feedback. In the early stages of continual learning, the incremental model may not have acquired sufficient knowledge, resulting in suboptimal performance on earlier subjects. This can lead to the generation of inaccurate pseudo-labels due to significant inter-subject variability. However, **this issue does not affect the model's training for two primary reasons:**
>
> 1. **Retention of High-Quality Pseudo Labels:** We retain only high-quality pseudo labels by applying a confidence threshold $\xi_2$ for inclusion in the storage Spseudo​ for subsequent replay. If the model does not adapt effectively to earlier subjects and generates low-confidence pseudo labels, these samples are not saved in Spseudo​, thereby ensuring the integrity of the replay samples.
>
> 2. **Selective Replay Strategy:** We employ a selective replay strategy by sampling from both $S_{true}$​ and $S_{pseudo}$​ in an 8:2 ratio. This approach allows us to replay only a limited number of pseudo-labeled samples generated during the continual learning process, thereby enhancing the diversity of the replay samples. In other words, even if some low-quality pseudo-label samples are introduced, their overall impact on the replay samples remains minimal.
>
>
> We hope these clarifications will address your concern.

---

> ### Author Response · Authors · 2024-11-20
> **Response to Reviewer J7FP[5/N]**
>
> **Q8:** How is the plasticity of the incremental set evaluated?; Is there a specific incremental split for training and testing?
>
> **R8:** Thanks for your insightful question. For Q8.1, in our proposed UICL setting, the incremental model needs to continuously adapt to each unseen individual one by one. After each round of adaptation, we evaluate the model’s plasticity on the latest individual. For example, the initial model $M_0$ needs to adapt to the first individual in the continual flow, resulting in the incremental model $M_1$. We calculate the metrics(i.e., ACC. MF1) of the model $M_1$ on the first individual to measure its plasticity. Then the incremental model $M_1$ need to adapt to the second individual and so on. After the model has adapted to the entire incremental set, we calculate the average ACC/MF1 obtained from each instance as the final plasticity performance.
>
> For Q8.2, there is no specific incremental split. The model performs unsupervised adaptation on an incremental individual and then validates its plasticity on the same individual. **The detailed explanations of the UICL process, including how to evaluate stability and plasticity, are listed in Appendix C, Fig. 9 (page 16).**
>
> ---
>
> **Q9:** What is the total number of samples stored in the storage buffer for each individual? In addition, how are the samples of the target domain replaced in the memory?
>
> **R9:** Thanks for your concern. For detailed responses to these questions, please **refer to R1.**
>
> ---
>
> **We highly appreciate again for your constructive and insightful feedback. If you have any further concerns, we would be pleased to address them.**

---

> ### Author Response · Authors · 2024-11-27
> **A Kind Reminder to Reviewer J7FP**
>
> Dear Reviewer ZsrG,
>
> Thank you for your thoughtful comments and for the positive aspects you highlighted. We have carefully addressed the key concerns with additional experiments in our rebuttal:
>
> - **Compared with Memory Sampling Methods:** We have added a new comparative study with other popular memory sampling methods (e.g., FIFO, Reservoir Sampling, Uniform Random Sampling).
>
> - **Compared with Recent Continual EEG Decoding Method:** We have included a recent cross-subject EEG-based continual learning method, ReSNT, for comparison.
>
> - **Technical Details of KL-based Penalty:** We have added the further clarification of technical details of CEA module.
>
> - **Partition Study:** According to your request, we have added a partition study to evaluate the model's performance under different dataset partitions.
>
> - **Technical Details of SSL Process:** We have added the further clarification of technical details of SSL process.
>
> We sincerely look forward to your constructive feedback. Your previous suggestions have greatly enhanced the quality of our manuscript. We believe that ongoing communication between authors and reviewers is essential for fostering collaboration and promoting advancements in our field. By sharing insights and constructive critiques, we can collectively address challenges and explore new directions for research in EEG/BCI technologies.
>
> Thank you once again for your support and constructive feedback!

---

> ### Author Response · Authors · 2024-11-28
> **We Would Appreciate Your Response**
>
> Dear Reviewer J7FP,
>
> We hope this message finds you well.
>
> We would like to extend our sincere gratitude for the valuable feedback you provided on our manuscript. Your insights are greatly appreciated and have significantly contributed to our work.
>
> We would like to kindly remind you that it has been over a week since we submitted our rebuttal. We are eager to know if our responses have adequately addressed your concerns. If there are any further issues or points you would like to discuss, we would be more than willing to clarify them during the remaining discussion phase.
>
> Thank you once again for your attention and support. We look forward to addressing any further questions you may have and refining our work based on your comments.
>
> Best regards,
>
> The authors

---

> ### Author Response · Authors · 2024-11-30
> **Invitation for the Second Period of Discussion**
>
> Dear Reviewer J7FP,
>
> Thank you for your thorough review and insightful questions. We would like to remind you that the extended discussion period is nearing its end. In our first-round response, we provided detailed replies and a summary **addressing your concerns,** specifically regarding:
>
> - Compared with Memory Sampling Methods
>
> - Compared with Recent Continual EEG Decoding Method
>
> - Technical Details
>
> - Partition Study
>
> We sincerely hope you will **reconsider your score** based on our clarifications, as this is crucial for us. Notably, several reviewers have already increased their scores or confidence following our explanations. If you have any further concerns, please feel free to reach out, and we will gladly provide additional clarification.
>
> Thank you for your consideration.

---

> ### Author Response · Authors · 2024-12-02
>
> Dear Reviewer J7FP
>
> As today marks the final day for feedback on our manuscript, we wanted to kindly follow up regarding your evaluation, which currently reflects a borderline rejection.
>
> In our earlier responses, we believe we have addressed your concerns comprehensively. We are eager to know if there are any additional suggestions or specific points we could consider to enhance our manuscript further.
>
> We sincerely hope you might reconsider your score or provide us with further insights that could guide us in strengthening our work.
>
> Thank you for your time and consideration.
>
> Best regards
>
> The authors

---

### Official Review · Reviewer_BBh5 · 2024-11-03

**Soundness:** 3
**Presentation:** 2
**Contribution:** 3
**Rating:** 8
**Confidence:** 5

**Summary:**

The work proposes a Continual learning-based framework for addressing the need for robustness against user-specific variability in EEG-based BCIs. The model agnostic approach combines Unsupervised Domain adaptation with a Continual learning framework. 3 different tasks with public datasets are used for the benchmark. Evaluation metrics use incremental individual test sets to measure plasticity and a dataset for generalisation to measure the stability of the approach.

**Strengths:**

The work addresses the domain's appropriate needs in terms of user variability. The approach is well proposed and benchmarked, including metrics compared with relevant SOTA, ablation studies and computational costs.
The work is technically detailed with appendices and presented with fair clarity.

**Weaknesses:**

The method section could be better represented with additional labels to the stages in Figure 2 that include the three stages explained in the overview: 1) producing pseudo labels, 2) updating models, and 3) updating storage. It wasn't easy to follow the complete process, shifting across figures, the overview section, each BrainUICL subsection and the appendix.

It's not a weakness per se. While the work is novel in its approach, authors can be more specific in contributions about the novelty of the approach across application domains. It is understood that the approach combines previously known approaches in Unsupervised domain adaptation and continual learning with novelty to the strategies in updating the replay buffer and the training loss, including cross-epoch alignment, where the motivation is similar to EwC.

**Questions:**

Quoting the lines from authors: Plasticity (P) denotes the model’s adapting ability to newly emerging individuals, while Stability
(S) indicates the model’s generalization ability to unseen individuals (i.e., new subjects)
Stability refers to the ability to maintain performance on previously seen and unseen individuals, including catastrophic forgetting. The current quote may lead to a misunderstanding. How well does it retain the performance on the dataset used for the M0 model?

The authors mention as follows:
We first explore the concept of Unsupervised Individual Continual Learning(UICL) in EEG-related applications, which is well-suited to the real-world scenario.

Is the concept of UICL novel or has it been proposed earlier? It is not clear from the subsequent discussion in related works. How is it different from Unsupervised Domain Adaption and CL combination apart from defining an individual as a domain?

The concept of generating pseudo labels is not clear. Appendix B clarifies the SSL mechanism used for incremental subjects. However, post-training, how are the pseudo-label confidence values generated, and how is the confidence threshold decided is not clear.


In section 3.3.2, the authors mention: "Here, we tend to utilize the real labeled samples for replay rather than the previously preserved
pseudo-labeled samples." Does this mean that the approach uses real labels for the selected pseudo-labeled samples?


Algorithm 1 on page 6 mentions Mg and Mi-1. However, while using DCB and CEA, Mg is not used and instead, Mi-1 is used. At the same time, the text mentions the use of CPC for adapting to the user's domain. Can the authors clarify this?

The authors do not mention the data preparation step for each dataset, i.e. how long the epochs are, any overlaps between the epochs, and details on the block sizes of the CNN. Some of these parameter choices are significant in evaluating the effectiveness and explainability of the approach.

The results reported in Table 3 and Figure 4 caption mention: Notably, all methods have five same input orders, and these orders are randomly different. It is unclear if the individuals added to the model are in the same order for each iteration. And are they shuffled randomly across those five iterations? I assume that the 95% CIs and SDs in Table 3 are coming from these 5 iterations of different orders.

Are the ACC and MF1 values averaged across incremental individuals with models Mi and across the five iterations of the order? The results are not clear after reading through the sections and looking at tabular data.


In Table 4, Figure 5, ablation results, it is surprising that the base performance(AAA and AAF1) does not decline with the addition of individuals. Does the base model have any replay? It would be good if authors could point to the section if already addressed.

---

> ### Author Response · Authors · 2024-11-20
> **Response to Reviewer BBh5[1/N]**
>
> **We'd like to express our sincere gratitude for your careful readings and valuable comments. We are glad to see you approved the contributions of our work.** In this rebuttal, we aim to address each of the key concerns and points you have raised.
>
> ---
>
> **Q1:** The method section could be better represented with additional labels to the stages in Figure 2 that include the three stages explained in the overview: 1) producing pseudo labels, 2) updating models, and 3) updating storage.
>
> **R1:** Thank you for bringing this to our attention. We have made modifications to the corresponding sections of Figure 2 to enhance the clarity of our approach. **The revised figure can be found in the newly uploaded file.**
>
> ---
>
> **Q2:** While the work is novel in its approach, authors can be more specific in contributions about the novelty of the approach across application domains.
>
> **R2:** Thank you for your valuable comment. We have revised a portion of the Introduction to emphasize our contributions, highlighting the following points:
>
> 1. **The Contribution to EEG-based Applications (page 2):** The proposed BrainUICL is well-suited to real-world scenarios where a large number of unseen and unordered individuals continuously emerge. It can not only enable the model to continuously adapt to a long-term individual flow in a plug-and-play manner, but also balancing the SP dilemma during such CL process.
> 2. **The Contribution to Technological Innovation (page 2):** We have designed two novel modules: the Dynamic Confident Buffer (DCB) and Cross Epoch Alignment (CEA) to tackle the aforementioned challenges. Specifically, the DCB employs a selective replay strategy that ensures the accuracy of labels for replay samples in an unsupervised setting while maintaining the diversity of these samples. The CEA module innovatively aligns the incremental model across different time states to prevent overfitting, ensuring that the incremental model remains unaffected by varying learning trajectories, which is particularly relevant given that continual flows are unordered in real-world scenarios.
>
> We hope that these points clarify our contributions. For further details, please refer to the newly uploaded file, where the modifications are highlighted in blue font. Thank you again for your valuable comments.
>
> ---
>
> **Q3:** Stability refers to the ability to maintain performance on previously seen and unseen individuals, including catastrophic forgetting. The current quote may lead to a misunderstanding.
>
> **R3:** Thank you for pointing this out. We agree and have revised the corresponding quote to avoid any misunderstanding.
>
> - Plasticity (P) denotes the model's ability to adapt to newly emerging individuals, while Stability (S) indicates the model's generalization ability to **both previously seen and unseen** individuals (i.e., new subjects) (**page 1**).
>
> Notably, we consider the model's generalization performance on unseen subjects as the primary measure of its stability. The rationale for this is as follows:
>
> Unlike other task scenarios (e.g., incremental learning in image classification), where the incremental model must adapt to new tasks/domains while also maintaining performance on previous tasks/domains, in the context of EEG-based individual continual learning, we typically do not need to retest previously seen subjects. Therefore, **we place greater emphasis on the model's generalization ability with respect to unseen subjects rather than those previously encountered.**

---

> ### Author Response · Authors · 2024-11-20
> **Response to Reviewer BBh5[2/N]**
>
> **Q4:** How well does it retain the performance on the dataset used for the M0 model?
>
> **R4:** Thanks for your concern. In accordance with your suggestion, we assessed the performance variations of the pretraining set (i.e., the dataset used for $M_0$ model) throughout the continual learning process, as illustrated in the table below:
>
> | Dataset | ACC ($M_0$) | ACC ($M_{N_T}$) | MF1 ($M_0$) | MF1 ($M_{N_T}$) |
> | --- | --- | --- | --- | --- |
> | ISRUC | 74.5 | 89.0 | 73.4 | 88.0 |
> | FACED | 38.1 | 99.6 | 34.0 | 99.6 |
> | Physionet-MI | 99.8 | 99.9 | 99.8 | 99.9 |
>
> Here, $M_0$ denotes the initial model and $M_{N_T}$ denotes the final model after continual adaptation to all incremental individuals. For the detailed performance variation curves, please refer to the Appendix. H, Fig. 10 (**page 19**). The results indicate that on the ISRUC and FACE datasets, the model's performance on the training set exhibits an overall improvement, rather than the catastrophic forgetting typically associated with continual learning. This is reasonable, considering that 80% of the replay samples during each iteration are sourced from the training set, thereby enhancing performance as we continuously replay the labeled samples from the training set.
>
> In our setup, the train set is used solely for pretraining the model $M_0$​ and does not participate in the subsequent continual learning process. **We place greater emphasis on the model's generalization ability concerning unseen subjects rather than those previously encountered** for the following reasons:
>
> - As mentioned in **R3**, in reality, the continual individual flow maintains a positive trajectory (**from past to future**), where unseen individuals arrive for adaptation and subsequently exit after the adaptation process. Therefore, the incremental model is typically not required to retest individuals who have already been adapted. If previously adapted subjects reappear, we can treat them as newly emerged individuals and have the model readapt to them.
>
> We hope this additional experiment will address your question. Thank you once again for your insightful feedback.
>
> ---
>
> **Q5: Is the concept of UICL novel or has it been proposed earlier? It is not clear from the subsequent discussion in related works.**
>
> **R5:** Thank you for pointing this out. Indeed, there are existing studies that propose cross-subject continual learning approaches in the EEG field [1-3] and address the issue of online EEG sequential decoding (we have added these new related works in the Related Work section). However, these studies have several limitations, which are outlined as follows:
>
> 1. **Limitation in Supervised Learning:** These studies are based on supervised learning, where the labels for incremental individuals are available. However, in real-world scenarios, the labels for newly emerging incremental individuals are often unknown.
> 2. **Limitation in their Evaluated Datasets:** These studies have been validated primarily on small-scale datasets, such as BCI IV-2a, DEAP, and SEED, which involve only a limited number of subjects. This limitation results in a short duration for the continual individual flow, making it challenging to effectively assess the stability and plasticity of incremental models in a long-term continual learning scenarios.
>
> To the best of our knowledge, we are the first to explore the concept of **Unsupervised Individual Continual Learning**, which is particularly well-suited for real-world scenarios where labels for incremental individuals are unavailable. Moreover, we have conducted our study on large-scale datasets comprising at least 100 subjects, enabling us to evaluate the model's stability and plasticity during long-term continual individual flows. We hope these points clarify the novelty of our proposed UICL paradigm. Thank you once again for your valuable feedback.
>
> [1] Online continual decoding of streaming EEG signal with a balanced and informative memory buffer[J]. Neural Networks, 2024, 176: 106338.
>
> [2] Replay with Stochastic Neural Transformation for Online Continual EEG Classification[C]//2023 IEEE International Conference on Bioinformatics and Biomedicine (BIBM). IEEE, 2023: 1874-1879.
>
> [3] Retain and Adapt: Online Sequential EEG Classification with Subject Shift[J]. IEEE Transactions on Artificial Intelligence, 2024.

---

> ### Author Response · Authors · 2024-11-20
> **Response to Reviewer BBh5[3/N]**
>
> **Q6:** How is it different from Unsupervised Domain Adaption and CL combination apart from defining an individual as a domain?
>
> **R6:** Thanks for your insightful question. The integration of Unsupervised Domain Adaptation (UDA) and Continual Learning (CL) can be summarized as Unsupervised Continual Domain Adaptation (UCDA). However, our work differs significantly from existing UCDA studies for several reasons:
>
> 1. **Difference in the Number of Incremental Domains:** Traditional UCDA-based scenario often faces limited incremental domains (e.g., style transfer increments, as the incremental types of styles are limited). However, in real-world scenarios, the emergence of new individuals is continuous and ongoing, leading to a long-term individual continual flow (i.e., domains). The model is required to have the ability to adapt to an exceptionally long continual flow and remain unaffected during long-term training.
>
> 2. **Difference in the Impact of Learning Trajectories:** Traditional UCDA research typically overlook the influence of continual flows with different input orders on the model. The effect of varying input orders on the learning trajectory is minimal in the context of limited incremental target domains.
>
>   However, in the real-world scenarios, there are numerous incremental individuals, and they appear in a completely unordered and continual flow. In this context, the impact of varying input orders within continual individual flows on the model's learning trajectory is significant, especially when the model encounters outliers characterized by markedly abnormal EEG signals during the early stages of the CL process. Such instances can lead to considerable deviations in the model's original learning trajectory, often resulting in a decline in performance that may be irreversible.
>
>
> Our method is capable of handling such long-term individual continual learning and remaining unaffected by outliers under different learning trajectories, meeting the practical needs in real life. We hope these points will provide a more comprehensive understanding of the novelty of our work.
>
> ---
>
> **Q7:** The concept of generating pseudo labels is not clear. Appendix B clarifies the SSL mechanism used for incremental subjects. However, post-training, how are the pseudo-label confidence values generated, and how is the confidence threshold decided is not clear.
>
> **R7:** Many thanks for your valuable concern. We have included a detailed description of the SSL mechanism in Appendix B (**page 15**), which covers the process of generating pseudo label confidence values, the generation of pseudo labels, and the criteria for selecting the confidence threshold. The details are as follows:
>
> 1. **Generating Pseudo Labels:** When an incremental individual arrives, we first apply the CPC algorithm to the guiding model $M_g$​, which is a copy of the most recent model $M_{i−1}$​, using the samples from the incremental individual. After adaptation, we utilize the fine-tuned guiding model​ to generate pseudo labels for subsequent training. Specifically, we obtain classification prediction probabilities (i.e., confidence values) for each sample by inputting the incremental individual samples into the guiding model $M_g$​ after the softmax layer. We then retain only those high-confidence pseudo labels with prediction probabilities exceeding the threshold $\xi_1$​ (0.90) for further training.
>
> 2. **Selecting the Confidence Threshold:** For the threshold $\xi_1$​, setting it too high may result in an insufficient number of generated pseudo labels, while setting it too low can introduce additional low-quality pseudo labels. To address this issue, we conducted a parameter selection experiment to evaluate the impact of different thresholds (0.75, 0.80, 0.85, 0.90, 0.95) on the performance of the generated pseudo labels. The experimental results indicate that the optimal performance is achieved when the confidence threshold $\xi_1$ is set to 0.90.
>
>
> We hope these additional clarifications will address your concerns.

---

> ### Author Response · Authors · 2024-11-20
> **Response to Reviewer BBh5[4/N]**
>
> **Q8:** In section 3.3.2, the authors mention: "Here, we tend to utilize the real labeled samples for replay rather than the previously preserved pseudo-labeled samples." Does this mean that the approach uses real labels for the selected pseudo-labeled samples?
>
> **R8:** Thanks for your concern. We apologize for this quote as it may lead to some misunderstanding. **We have revised the corresponding quote to avoid any misunderstanding.**
>
> -  we utilize relatively more real labeled samples from the $S_{true}$, and relatively less previously preserved pseudo-labeled samples from the $S_{pseudo}$ for replay (**page 5**).
>
> In DCB module, we replay more real samples from the training set to ensure the accuracy of the labels for the replay samples. Meanwhile, we replay a small amount of pseudo-labeled samples produced from the CL process to increase the diversity of the replay samples. Specifically, in our DCB module, at each time step, we select buffer samples from both $S_{true}$ and $S_{pseudo}$ in an 8:2 ratio.
>
> ---
>
> **Q9:** Algorithm 1 on page 6 mentions Mg and Mi-1. However, while using DCB and CEA, Mg is not used and instead, Mi-1 is used. At the same time, the text mentions the use of CPC for adapting to the user's domain. Can the authors clarify this?
>
> **R9:** Many thanks for your concern. For detailed process of SSL, please refer to the **R7, Generating Pseudo Label**. Each adaptation of the guiding model $M_g$ based on the incremental individual is solely intended to provide high-confidence pseudo labels for the subsequent training of the incremental model $M_{i-1}$​. The guiding model $M_g$​ itself does not participate in the subsequent training (i.e., DCB, CEA).
>
> ---
>
> **Q10:** The authors do not mention the data preparation step for each dataset, i.e. how long the epochs are, any overlaps between the epochs, and details on the block sizes of the CNN. Some of these parameter choices are significant in evaluating the effectiveness and explainability of the approach.
>
> **R10:** Thanks for your valuable and helpful suggestions. We apologize for missing these specific details. We have added the missing details as follows:
>
> **Data preparation:**
>
> **ISRUC:** A sleep dataset consisted of the three sub-groups. We specifically selected sub-group 1, which consists of all-night polysomnography (PSG) recordings from 100 adult individuals and contains 86400 samples. We use six EEG channels (F3-A2, C3-A2, O1-A2, F4-A1, C4-A1, O2-A1) and two EOG channels (E1-M2, E2-M1), and the data is resampled to 100 Hz for evaluation. All EEG signals are divided into 30-second segments, which are then categorized into five distinct sleep stages (Wake, N1, N2, N3, REM) by sleep experts based on the standards set by the American Academy of Sleep Medicine (AASM)[1]. The transition patterns between sleep epochs are essential for sleep staging. In line with previous sleep staging studies[2], we treat this task as a sequence-to-sequence classification problem, defining the sequence length as 20, which corresponds to one sleep sequence consisting of 20 30-seconds samples. We excluded subject 8 and 40 due to some missing channels.
>
> **FACED:** A large finer-grained affective computing EEG dataset covers nine emotion categories (amusement, inspiration, joy, tenderness, anger, fear, disgust, sadness, and neutral emotion) from recordings of 123 subjects. Each recording contains 32-channel EEG signals at 250 Hz sampling rate. All EEG signals are divided into 10-second segments. All the 123 recordings were used for evaluation.
>
> **Physionet-MI:** A motor imagery EEG dataset covers four motor classes (left fist, right fist, both fists and both feet) from recordings of 109 subjects. Each recording contains 64-channel EEG signals at 160 Hz sampling rate. All EEG signals are divided into 4-second segments. All the 109 recordings were used for evaluation.
>
> We have added the detailed data preparation in the Appendix. D (**page 16**). And **the details of the CNN block have been supplemented in the Appendix. D, Tab. 7 (page 18).**
>
> [1] The American Academy of Sleep Medicine (AASM) Manual for the Scoring of Sleep and Associated Events: Rules, Terminology and Technical Specifications, volume 1. American academy of sleep medicine Westchester,IL, 2007.
>
> [2] Automatic sleep staging of eeg signals: recent development, challenges, and future directions. Physiological Measurement, 2022.

---

> ### Author Response · Authors · 2024-11-20
> **Response to Reviewer BBh5[5/N]**
>
> **Q11:** The results reported in Table 3 and Figure 4 caption mention: Notably, all methods have five same input orders, and these orders are randomly different. It is unclear if the individuals added to the model are in the same order for each iteration. And are they shuffled randomly across those five iterations?
>
> **R11:** Thanks for your valuable concern. **The five input orders are different and generated by randomly shuffling the data for statistical evaluation.** The order of individuals in the continual flow is completely randomized based on the initial random seeds. Notably, in our study, while maintaining consistent dataset partitioning, we only altered the input order of the continual individual flow (by changing the initial random seed) to assess the impact of different input orders (i.e., learning trajectories) on the model, repeating this process five times in total. To facilitate understanding, we provide a simple illustrative example, as shown in the table below:
>
> |     | Train Set | Generalization Set | Incremental Set (i.e., Continual Individual Flow) |
> | --- | --- | --- | --- |
> | **Order 1** | 1, 2, 3 | 4, 5 | 6 -> 7 -> 8 -> 9 -> 10 |
> | **Order 2** | 1, 2, 3 | 4, 5 | 8 -> 9 -> 6 -> 7 -> 10 |
> | **Order 3** | 1, 2, 3 | 4, 5 | 10 -> 9 -> 6 -> 8 -> 7 |
> | **Order 4** | 1, 2, 3 | 4, 5 | 9 -> 8 -> 6 -> 7 -> 10 |
> | **Order 5** | 1, 2, 3 | 4, 5 | 7 -> 9 -> 10 -> 8 -> 6 |
>
> Here, the numbers denote the different individual IDs. In Fig. 4 and Fig. 5, it shows how the different input orders affect different model's performance. The shaded areas indicate each method's 95% confidence intervals under different orders. The shaded area is larger, the influence of the different input orders greater. Influenced by varying learning trajectories, some comparative methods show significant performance gaps. **In comparison, our model remains largely unaffected by learning trajectories. This characteristic is particularly well-suited for real-world scenarios, where the emergence of incremental individuals is entirely unordered and unknown.** We have added the detailed explanations in the Appendix. C. (**page16**)
>
> ---
>
> **Q12:** Are the ACC and MF1 values averaged across incremental individuals with models Mi and across the five iterations of the order? The results are not clear after reading through the sections and looking at tabular data.
>
> **R12:** Thanks for your valuable question. Yes, for each input order(i.e., iteration), we calculate the average ACC and average MF1 across all the incremental individuals. After five iterations, we calculate the average of the average results(i.e., average ACC and average MF1) from each iteration to provide a statistical results. **We have modified the original text to make it easier to understand (page 7).**

---

> ### Author Response · Authors · 2024-11-20
> **Response to Reviewer BBh5[6/N]**
>
> **Q13:** In Table 4, Figure 5, ablation results, it is surprising that the base performance(AAA and AAF1) does not decline with the addition of individuals. Does the base model have any replay? It would be good if authors could point to the section if already addressed.
>
> **R13:** Thanks for your insightful question. Our base model employs a uniform random strategy, wherein all incoming batch samples are stored in memory. Each time, we randomly select samples from the storage to fill the replay buffer.
>
> The base model's performance does not experience a significant decline with the addition of individuals, because we introduce a hyper-parameter $\alpha$ which regulates the influence of the new incoming individuals on the model performance. The $\alpha$ is designed in the loss function (Eq. 4, **page 6**), and all the methods in the ablation study use this loss function. Specifically, as the continual learning process advances, α gradually decreases, while the penalty imposed on incremental individuals correspondingly increases. This approach ensures that the model's performance is progressively less affected by incremental individuals, promoting stability over time. This explains why the base model does not experience a significant decline in performance during the later stages of training (i.e., its performance improves relative to the initial performance).
>
> However, even with the assistance of α, experimental results indicate that the base model still encounters the following issues in the absence of the DCB and CEA modules:
>
> 1. **Performance Decline in the Later Stages of Continual Learning:** As illustrated in Fig. 5 (**page 10**), on the ISRUC and Physionet-MI datasets, the base model experiences a continuous decline in performance during the later stages of continual learning, which is particularly pronounced in the Physionet-MI dataset. While there is still an improvement in performance compared to the initial state (i.e., performance of $M_0$​), the base model exhibits a downward trend over time in subsequent learning phases.
> 2. **Instability under Different Learning Trajectories:** As clearly illustrated in Fig. 5, the area of the 95% confidence interval for the base model (represented by the shaded region in the figure) is significantly larger than that of the other ablated methods, exhibiting a divergent trend. This suggests that, in the absence of the DCB and CEA modules, the base model is highly sensitive to variations in the input order of different individual flows.
>
> In comparison to these ablated methods, the performance of our approach not only increases progressively over time, but the confidence intervals also tend to converge. This demonstrates the effectiveness of our method in handling long-term individual continual learning. We hope the provided explanations will address your concerns. Thank you once again for your valuable feedback.
>
> ---
>
> **Great appreciation for your encouraging comments. Your constructive and insightful feedback has improved the quality of our paper.**

---

> > ### Comment · Reviewer_BBh5 · 2024-11-26
> >
> > Thank the authors for the clarifications and additional details. More confident with the review score post the reply.

---

> > > ### Author Response · Authors · 2024-11-26
> > > **Thanks for your engagement**
> > >
> > > Thank you very much for maintaining the positive score! We are grateful for your attentive and constructive feedback.

---

### Official Review · Reviewer_bzyV · 2024-11-03

**Soundness:** 1
**Presentation:** 2
**Contribution:** 1
**Rating:** 1
**Confidence:** 5

**Summary:**

Individual differences are evident in EEG datasets, and the authors employed continuous learning to facilitate adaptive models for handling new subjects or patients.

**Strengths:**

Authors tried to use continual learning to adaptively manage individual differences in EEG signals.

**Weaknesses:**

The claim regarding this study’s contribution is confusing, and the related work review is limited. Individual differences in EEG data are a well-known challenge, and substantial prior work in supervised learning and transfer learning has effectively addressed this issue using robust feature representations. There are many popular EEG datasets for classification tasks that were not discussed and considered.

The authors argue that existing EEG models lack practical applicability, especially in clinical settings with diverse patient profiles (refer to abstract). However, their selected EEG datasets do not include patient data, covering only sleep, emotion, and motor imagery tasks—none involving clinical data. Moreover, several widely-used EEG datasets for classification tasks are notably absent from their analysis.

Previous work on the datasets (above mentioned) they examined has achieved over 90% accuracy in classification tasks through supervised or transfer learning, which suggests these approaches can manage individual differences well. In contrast, this study reports accuracy levels around 40%, which raises the question: what factors account for this significant performance gap?

The role of cross-epoch alignment is unclear, particularly regarding its effectiveness in managing within- and across-subject variations. A more detailed explanation of its purpose and impact on these aspects is needed.

**Questions:**

See above.

---

> ### Author Response · Authors · 2024-11-20
> **Response to Reviewer bzyV[1/N]**
>
> **We would like to express our sincere gratitude to you for taking the time to review our submission.** In this rebuttal, we will address each of the key issues and points you have raised.
>
> ---
>
> **Q1:** The claim regarding this study’s contribution is confusing, and the related work review is limited.
>
> **R1:** Thanks for your valuable concerns. **We'd like to address your concerns from the following two perspectives.**
>
> 1. **Emphasizing the contributions:**
>
>   - **The Contribution to EEG-based Applications:** The proposed BrainUICL is well-suited to real-world scenarios where a large number of unseen and unordered individuals continuously emerge. It can not only enable the model to continuously adapt to a long-term individual flow in a plug-and-play manner, but also address the issue of individual differences.
>
>   - **The Contribution to Technological Innovation:** To address the challenge of managing long-term and unordered individual flows in a continual learning framework, we have designed two novel modules: the Dynamic Confident Buffer (DCB) and Cross Epoch Alignment (CEA). Specifically, the DCB employs a selective replay strategy that ensures the accuracy of labels for replay samples in an unsupervised setting while maintaining the diversity of these samples. The CEA module innovatively aligns the incremental model across different time states to prevent overfitting, ensuring that the incremental model remains unaffected by varying learning trajectories, which is particularly relevant given that continual flows are unordered in real-world scenarios.
>
> 2. **Addition of Related Works:** We have added citations[1-3] for the previously missing works (**page 3**) and rewritten the related work section according to the following structure: EEG Decoding, Continual Learning, and Continual EEG Decoding. We reorganized the "continual learning" to the regularization based methods, the parameter isolation based methods and the rehearsal based methods. Meanwhile, we distinguish the continual EEG decoding from the classic EEG decoding, and introduce how continual learning works for the EEG analysis.
>
>
> We hope that these points clarify our contributions and that the additional related work provides a more comprehensive overview of EEG-based continual learning efforts. For further details, please refer to the newly uploaded file, where the modifications are highlighted in blue font. Thank you again for your valuable comments.
>
> [1] Online continual decoding of streaming EEG signal with a balanced and informative memory buffer[J]. Neural Networks, 2024, 176: 106338.
>
> [2] Replay with Stochastic Neural Transformation for Online Continual EEG Classification[C]//2023 IEEE International Conference on Bioinformatics and Biomedicine (BIBM). IEEE, 2023: 1874-1879.
>
> [3] Retain and Adapt: Online Sequential EEG Classification with Subject Shift[J]. IEEE Transactions on Artificial Intelligence, 2024.

---

> ### Author Response · Authors · 2024-11-20
> **Response to Reviewer bzyV[2/N]**
>
> **Q2:** Individual differences in EEG data are a well-known challenge, and substantial prior work in supervised learning and transfer learning has effectively addressed this issue using robust feature representations.
>
> **R2:** Thanks for your concern. Our task focuses on individual continual learning in real-world scenarios. This setting presents two primary challenges:
>
> - **The continuously emerging new individuals are unknown (without labels)**
> - **The emergence of new individuals is unordered and random, which necessitates adaptation in a plug-and-play manner.**
>
> These challenges cannot be addressed by traditional supervised learning or transfer learning methods with the following reasons:
>
> 1. **Limitation in Supervised Learning:** For supervised methods, the primary issue is that, in practice, we cannot obtain the labels for unknown subjects in advance. In other words, **we need a unsupervised fine-tuning of the model on newly emerging unknown subjects.**
>
> 2. **Limitation in Transfer Learning:** For transfer learning methods, existing unsupervised domain adaptation (UDA) techniques can effectively address individual discrepancies between the source and target domains. However, this approach presents the challenge in real-world scenarios. Since most UDA methods treat the target domain as a whole (i.e., multiple individuals), necessitating the availability of a batch of target domain samples before adaptation can occur. This is impractical in real life, where the arrival of each new individual is entirely random. **We need a plug-and-play adaptation approach rather than waiting for all target individuals to arrive before conducting the adaptation.**
>
>
> To address these challenges, the optimal approach is to employ an incremental model that can continuously adapt to all newly emerged unknown individuals in a plug-and-play manner. **The proposed BrainUICL is well-suited for real-world scenarios, as the pre-trained model is capable of continuously adapting to newly appeared unknown individuals at any time during daily life.**
>
> We hope these points offer a clearer understanding of the significance of our work and address your concerns.
>
> ---
>
> **Q3:** There are many popular EEG datasets for classification tasks that were not discussed and considered.
>
> **R3:** Thanks for your question. We appreciate your feedback on our dataset selection. Below are our responses concerning the datasets we evaluated：
>
> The advantage of our framework lies in its **long-term individual continual adaptation**, meeting the requirements of real-world scenarios where a large number of unseen individuals continuously arrive. To evaluate our framework, we need relatively large datasets which can closely simulate the long and continual data flow in real-world scenarios. Therefore, we selected three large datasets composed of at least 100 subjects for evaluation. We did not choose some other mainstream datasets, due to their small number of subjects (e.g., DEAP[1] with only 32 subject, SEED[2] with only 15 subjects, CHB-MIT[3] with only 23 subjects). Furthermore, we believe that **the datasets we selected include both resting state and task state EEG signals, as well as data from both healthy individuals and patients, providing sufficient diversity to validate the performance of our model.**
>
> We hope these clarifications address your concerns about the datasets selection. Thank you again for your valuable feedback.
>
> [1] Koelstra S, Muhl C, Soleymani M, et al. Deap: A database for emotion analysis; using physiological signals[J]. IEEE transactions on affective computing, 2011, 3(1): 18-31.
>
> [2]Zheng W L, Lu B L. Investigating critical frequency bands and channels for EEG-based emotion recognition with deep neural networks[J]. IEEE Transactions on autonomous mental development, 2015, 7(3): 162-175.
>
> [3] PhysioBank, PhysioToolkit, and PhysioNet: components of a new research resource for complex physiologic signals[J]. circulation, 2000, 101(23): e215-e220.

---

> ### Author Response · Authors · 2024-11-20
> **Response to Reviewer bzyV[3/N]**
>
> **Q4:** The authors argue that existing EEG models lack practical applicability, especially in clinical settings with diverse patient profiles (refer to abstract). However, their selected EEG datasets do not include patient data, covering only sleep, emotion, and motor imagery tasks—none involving clinical data. Moreover, several widely-used EEG datasets for classification tasks are notably absent from their analysis.
>
> **R4:** Thanks for your concern. There may be some misunderstanding. **The selected EEG datasets have included patient data (i.e., ISRUC group1).** The following is a quote from the original paper of ISRUC[1]:
>
> - "Subgruop-Ⅰ: Data of 100 adult subjects with evidence of having **sleep disorders**."
>
> It is well known that the EEG signals of patients exhibit more significant differences compared to those of healthy individuals. Our experimental results indicate that our method not only works effectively on healthy individuals but also demonstrates good performance on datasets composed of patients (**pages 8-9**).
>
> In response to the question, "Moreover, several widely used EEG datasets for classification tasks are notably absent from their analysis," please refer to **R3**. We hope that the explanations regarding the selected datasets could address your concerns.
>
>
> ---
>
> **Q5:** Previous work on the datasets (above mentioned) they examined has achieved over 90% accuracy in classification tasks through supervised or transfer learning, which suggests these approaches can manage individual differences well. In contrast, this study reports accuracy levels around 40%, which raises the question: what factors account for this significant performance gap?
>
> **R5:** Thanks for your valuable concern. Based on your statement, we assume that the dataset you are referring to is Physionet-MI (as the other two datasets do not match your description). There may be some misunderstanding about this performance gap with the following reasons:
>
> 1. **Physionet-MI can be used for four/binary Classification:** The work **you mentioned which achieves 90% accuracy, is based on binary classification[2, 3]**. In contrast, **our evaluation includes all four classes,** which introduces significantly greater complexity to the classification task. Physionet-MI includes four classes (**left fist, right fist, both fists and both feet**). Additionally, it can also be used for binary classification tasks (**left fist, right fist**). Here are some quotes from these original paper:
>
>   - **EEGSym[2] (Acc: 88.6±9.0):**"The imagination consisted of opening/closing either the left or right hand."
>
>   - **Georgios. et al[3] (Acc: 86.36)** "We choose to work on the two-class problem of classifying left-hand versus right-hand imaginary movements, discarding the data from the other classes."
>
> 2. **Different Dataset Partition:** our dataset partitioning method is quite different from that of previous studies. For example, EEGSym[2] employs **LOSO (leave-one-subject-out)** for evaluation, which means they use data from 108 subjects to pretrain a model and test on the last one. In contrast, in our setting, the dataset is divided into three parts: pretraining, incremental, and generalization sets. **We only use a small amount of labeled data to pretraining**, and then the pretrained model adapts to the incremental individuals one by one.
>
>
> It is reasonable to anticipate that **the classification accuracy for a four-class problem will be significantly lower than that for a binary classification task, particularly given that we utilized only a limited amount of data for pre-training instead of a substantial dataset.** We hope this explanation provides clarity and context for our reported results.
>
> [1] Khalighi S, Sousa T, Santos J M, et al. ISRUC-Sleep: A comprehensive public dataset for sleep researchers[J]. Computer methods and programs in biomedicine, 2016, 124: 180-192.
>
> [2] Pérez-Velasco S, Santamaría-Vázquez E, Martínez-Cagigal V, et al. EEGSym: Overcoming inter-subject variability in motor imagery based BCIs with deep learning[J]. IEEE Transactions on Neural Systems and Rehabilitation Engineering, 2022, 30: 1766-1775.
>
> [3] Zoumpourlis G, Patras I. Motor imagery decoding using ensemble curriculum learning and collaborative training[C]//2024 12th International Winter Conference on Brain-Computer Interface (BCI). IEEE, 2024: 1-8.

---

> ### Author Response · Authors · 2024-11-20
> **Response to Reviewer bzyV[4/N]**
>
> **Q6:** The role of cross-epoch alignment is unclear, particularly regarding its effectiveness in managing within- and across-subject variations. A more detailed explanation of its purpose and impact on these aspects is needed.
>
> **R6:** Many thanks for your insightful comment. The core idea of BrainUICL is to impose a penalty on incremental individuals to prevent the model from overfitting to them and forgetting previously acquired knowledge. Accordingly, we propose the Cross Epoch Alignment (CEA) module to implement a soft penalty on incremental individuals. Specifically, we align the distribution of the previous model states every two epochs. When the model begins to overfit to new individuals, this is mitigated by aligning with the distribution of earlier model states. This approach is beneficial as it effectively prevents the model from overfitting to specific individuals(**especially outliers**, this part of analyse is listed in Appendix. G, **page 19**), **thereby avoiding a deviation from the original learning trajectory and ensuring the model stability during such long-term continual learning process.** Furthermore, we conducted a study to assess the impact of different selections for the alignment interval (see Appendix. E.2, **page 17**). The results indicate that the performance is optimal when the alignment is operated every two epochs.
>
> ---
>
> **Thanks once again for taking the time to provide your valuable comments. If you have any further concerns, we would be pleased to address them.** For more detailed revisions to the article, please refer to the newly uploaded file, where we have made improvements in both the main text and the appendix. The modifications are highlighted in blue font.

---

### Official Review · Reviewer_CyMS · 2024-11-05

**Soundness:** 3
**Presentation:** 2
**Contribution:** 3
**Rating:** 8
**Confidence:** 5

**Summary:**

The author proposed to address the problem that EEG-based model trained on fixed datasets cannot generalize well to the continual flow of numerous unseen subjects in real-world scenarios. The authors propose BrainUICL which enables the EEG-based model to continuously adapt to the incoming new subjects, involving the Dynamic Confident Buffer (DCB) to selectively review the past knowledge and Cross Epoch Alignment (CEA) method to align the model at different time states.

**Strengths:**

The work is tackling an important problem which potentially can have significant impact in real world. The manuscript is easy to follow in general and the method caters well to the problem settings.

**Weaknesses:**

It is recommended that the authors to test the model on a wider range of EEG datasets covering different tasks for evaluation of model effectiveness, such as DEAP and high gamma etc.

Detailed analysis on memory cost is needed for the proposed operations such as the dynamic confident buffer and the cross epoch alignment.

How the different individuals are ordered during the continual learning process? Are they ordered by id or other attributes? Would different ordering affect the model performance much?

Recent works that also cover the exact topic of continual learning on EEG signal are missing in related work section, such as [1][2][3].

I would recommend a more modulized fomulation of related works, e.g. explictly divide the continual learning approaches into subsections   such as regularization, memory based approaches etc., and also distinguish between classic EEG decoding with continual EEG decoding for the EEG analysis part.

Given the work tackles specifically the EEG signal related task, better to highlight in introduction of the possible impact for the proposed continual EEG learning algorithm in real world applications.

More detailed explanation is needed for figures in the manuscript such as Fig. 3, 5 etc.


[1] Online continual decoding of streaming EEG signal with a balanced and informative memory buffer, Neural Networks, 2024
[2] Replay with Stochastic Neural Transformation for Online Continual EEG Classification, BIBM 2023
[3] Retain and Adapt: Online Sequential EEG Classification with Subject Shift, IEEE Transactions on Artificial Intelligence, 2024

**Questions:**

As listed in the strength and weaknesses sections above.

---

> ### Author Response · Authors · 2024-11-20
> **Response to Reviewer CyMs[1/N]**
>
> **Many thanks for the your detailed and insightful suggestions. We are glad to see you approved the contributions of our work.** In this rebuttal, we aim to address each of the key issues and points you have raised.
>
> ---
>
> **Q1:** It is recommended that the authors to test the model on a wider range of EEG datasets covering different tasks for evaluation of model effectiveness, such as DEAP and high gamma etc.
>
> **R1:** Thanks for your valuable comment. We appreciate your feedback on our dataset selection. Below are our responses concerning the datasets we evaluated：
>
> The advantage of our framework lies in its **long-term individual continual adaptation**, meeting the requirements of real-world scenarios where a large number of unseen individuals continuously arrive. To evaluate our framework, we need relatively large datasets which can closely simulate the long and continual data flow in real-world scenario. There, we selected three large datasets composed of at least 100 subjects for evaluation. We did not choose some other mainstream datasets, due to their small number of subjects (e.g., DEAP[1] with only 32 subject, SEED[2] with only 15 subjects, CHB-MIT[3] with only 23 subjects).
>
> ---
>
> **Q2:** Detailed analysis on memory cost is needed for the proposed operations such as the dynamic confident buffer and the cross epoch alignment.
>
> **R2:** We appreciate the helpful comment. **In conclusion, the memory cost of DCB and CEA is quite low.** For DCB, whenever the model adapts to an incremental individual, we only save the high-confidence pseudo-label $\tilde{Y_T}$ from the sample-label pairs {$X_T$, $\tilde{Y_T}$} into the buffer storage. Since the corresponding samples $X_T$ have already been saved, we only need to record their addresses, which incurs a low memory cost. During each iteration, we select a small batch of buffer samples for replay, further reducing the memory footprint. For CEA, we just need to save the buffer feature $F_B$ produced by the current model every 2 epochs. The memory cost for saving such a small batch of buffer features is low.
>
> ---
>
> **Q3:** How the different individuals are ordered during the continual learning process? Are they ordered by id or other attributes? Would different ordering affect the model performance much?
>
> **R3:** Thanks for your concerns. **The order of individuals in the continual flow is completely randomly shuffled by the initial random seeds.** Notably, in our study, while ensuring consistent dataset partitioning, we randomly shuffled the input order of the continual individual flow (by changing the initial random seed), to investigate the impact of different input orders (i.e., learning trajectories) on the model's performance. This process was repeated five times in total.
>
> To facilitate understanding, we provide a simple specific example, as shown in the table below.
>
> |     | Train Set | Generalization Set | Incremental Set (i.e., Continual Individual Flow) |
> | --- | --- | --- | --- |
> | **Order 1** | 1, 2, 3 | 4, 5 | 6 -> 7 -> 8 -> 9 -> 10 |
> | **Order 2** | 1, 2, 3 | 4, 5 | 8 -> 9 -> 6 -> 7 -> 10 |
> | **Order 3** | 1, 2, 3 | 4, 5 | 10 -> 9 -> 6 -> 8 -> 7 |
> | **Order 4** | 1, 2, 3 | 4, 5 | 9 -> 8 -> 6 -> 7 -> 10 |
> | **Order 5** | 1, 2, 3 | 4, 5 | 7 -> 9 -> 10 -> 8 -> 6 |
>
> Here, the numbers denote the different individual IDs. In Fig. 4 and Fig. 5, it shows how the different input orders affect different model's performance. The shaded areas indicate each method's 95% confidence intervals under different orders. The shaded area is larger, the influence of the different input orders greater. Influenced by varying learning trajectories, some comparative methods show significant performance gaps. **In comparison, our model remains largely unaffected by learning trajectories. This characteristic is particularly well-suited for real-world scenarios, where the emergence of incremental individuals is entirely unordered and unknown.**
>
> [1] Deap: A database for emotion analysis; using physiological signals[J]. IEEE transactions on affective computing, 2011, 3(1): 18-31.
>
> [2] Investigating critical frequency bands and channels for EEG-based emotion recognition with deep neural networks[J]. IEEE Transactions on autonomous mental development, 2015, 7(3): 162-175.
>
> [3] PhysioBank, PhysioToolkit, and PhysioNet: components of a new research resource for complex physiologic signals[J]. circulation, 2000, 101(23): e215-e220.

---

> ### Author Response · Authors · 2024-11-20
> **Response to Reviewer CyMs[2/N]**
>
> **Q4:** Recent works that also cover the exact topic of continual learning on EEG signal are missing in related work section, such as [1][2][3]. I would recommend a more modulized fomulation of related works, e.g. explictly divide the continual learning approaches into subsections such as regularization, memory based approaches etc., and also distinguish between classic EEG decoding with continual EEG decoding for the EEG analysis part.
>
> **R4:** Many thanks for pointing out this. We fully agree with your suggestions. In accordance with the suggested revisions, we have made the following changes to the article:
>
> 1. **Addition of new Related Works:** We have added citations for the previously missing works (**page 3**) and rewritten the related work section according to the following structure: EEG Decoding, Continual Learning, and Continual EEG Decoding. We reorganized the "continual learning" to the regularization based methods, the parameter isolation based methods and the rehearsal based methods. Meanwhile, we distinguish the continual EEG decoding from the classic EEG decoding, and introduce how continual learning works for the EEG analysis.
>
> 2. **Addition of new Comparative Method:** We have implemented the ReSNT [2] and compared it with our model (**page 9**). Since ReSNT is a supervised continual learning method, we made modifications during the reproduction process to enable it to function within our proposed unsupervised individual continual learning framework. Specifically, when an incremental individual arrives, we apply our SSL method (i.e., CPC) to generate high-confidence pseudo-labels for subsequent supervised fine-tuning of ReSNT. We conducted a statistical evaluation of the ReSNT on all the datasets, shown in Tab. 3 (**page 9**). Our method still outperform it.
>
>
> We hope these changes will address your concerns. Thank you once again for your valuable suggestions. **All the revisions can be seen in the newly uploaded file.**
>
> ---
>
> **Q5**: Given the work tackles specifically the EEG signal related task, better to highlight in introduction of the possible impact for the proposed continual EEG learning algorithm in real world applications.
>
> **R5:** Thank you for your insightful feedback. We have revised a portion of the Introduction (**Page 2**) to emphasize the significance of our work in real-world scenarios. The additional text is presented as follows:
>
> -   "It is well-suited to real-world scenarios where a large number of unseen and unordered individuals continuously arrive, enabling the model to continuously adapt to a long-term individual flow in a plug-and-play manner, while also balancing the SP dilemma during such CL process."
>
>
> ---
>
> **Q6:** More detailed explanation is needed for figures in the manuscript such as Fig. 3, 5 etc.
>
> **R6:** Thank you for bringing this to our attention. We have provided detailed explanations for the mentioned figures in the newly uploaded file. The specific revisions are as follows:
>
> 1. Fig. 3 caption (**page 6**): "The hyper-parameter $\alpha$ controls the influence of incremental individuals on the model. As $\alpha$ decreases throughout the continual learning process, the impact of incremental individuals on the model decreases."
>
> 2. Fig. 5 caption (**page 10**): "AAA and AAF1 curves of the ablated methods. Each point denotes an individual from the continual individual flow with the middle-line indicating the mean value of the AAA and AAF1 metrics under different input orders, while the shaded areas indicate their 95\% confidence intervals. Notably, all methods share five same input orders and these orders are randomly different. The experimental results demonstrate the effectiveness of the proposed DCB and CEA components."
>
>
> ---
>
> **We greatly appreciate your valuable comments. Your constructive feedback has significantly enhanced the quality of our paper.** For more detailed revisions to the article, please refer to the newly uploaded file, where we have made improvements in both the main text and the appendix. The modifications are highlighted in blue font.
>
> [1] Online continual decoding of streaming EEG signal with a balanced and informative memory buffer[J]. Neural Networks, 2024, 176: 106338.
>
> [2] Replay with Stochastic Neural Transformation for Online Continual EEG Classification[C]//2023 IEEE International Conference on Bioinformatics and Biomedicine (BIBM). IEEE, 2023: 1874-1879.
>
> [3] Retain and Adapt: Online Sequential EEG Classification with Subject Shift[J]. IEEE Transactions on Artificial Intelligence, 2024.

---

### Author Response · Authors · 2024-11-20
**Global Response and Revision of the Paper**

**We thank the reviewers for their insightful feedback.** We appreciate the positive comments on the noverlty of BrainUICL (CyMS, BBh5, J7FP), the impact on real-world scenarios(CyMS, BBh5, J7FP), the technological innovation(BBh5), the different evaluated datasets(BBh5, J7FP), the superior results(BBh5, J7FP), the clear presentation(CyMS, BBh5), the well proposed approach(BBh5).

We acknowledge some reviewers' concerns regarding the detailed analysis on memory cost, the detailed dataset partiton, some missing related works, the detailed explanation of the SSL, DCB and CEA modules, the choice of the datasets, the performance under different dataset partition and the detailed data preparation. **In this rebuttal, we have addressed the reviewers concerns through further comparative experiments, detailed technical explanations, and additional analysis.** This represents the best effort we can achieve within the limited timeframe allocated for the rebuttal.

We have revised our original manuscript. Below, we outline the specific revisions made in the updated version of our paper:

1. As reviewer BBh5 suggested, we have modified a quote in Introduction Section.**(Line47, Page1)**

2. In response to the suggestions of multiple reviewers, we have enhanced the description of our contribution.**(Line90-98, Page2)**

3. As reviewer BBh5 suggested, we have modified Figure 2 to provide a clearer description. **(Line108-123, Page3)**

4. As reviewer CyMS suggested, we have reorganized the related work.**(Line131-154, Page3)**

5. As reviewer BBh5 suggested, we have modified a quote in Methodology Section. **(Line259-263, Page5)**

6. As reviewer CyMs suggested, we have added detailed explanations in Fig.3 and Fig. 5.**(Line308-314, Page6; Line511-515, Page10)**

7. As reviewer BBh5 suggested, we have added the detailed explanation of the evaluating merics.**(Line365-369, Page7)**

8. In response to the suggestions of multiple reviewers, we have added a new comparative method.**(Line411-412, Line440, Page8-9)**

9. In response to the suggestions of multiple reviewers, we have added the details of the SSL process.**(Line778-834, Page15-16)**

10. As reviewer BBh5 suggested, we have added the section "Data Preparation" in the Appendix. D.**(Line863-891, Page16-17)**

11. In response to the suggestions of multiple reviewers, we have added the details of the DCB module.**(Line908-910, Page17)**

12. In response to the suggestions of multiple reviewers, we have added the details of the CEA module.**(Line914-942, Page17-18)**

13. As reviewer BBh5 suggested, we have added detailed parameters of the CNN blocks.**(Line952-955, Page18)**

14. We have removed the "Computational Cost" to the Appendix. F due to space constraints.**(Line967-980, Page18-19)**

15. As reviewer BBh5suggested, we have added the section"Performance Variation in Train Set" in the Appendix. H.**(Line1008-1025, Page19)**

16. As reviewer J7FP suggested, we have added the section"Compared with other Memory Sampling Methods" in the Appendix. I.**(Line1028-1071, Page19-20)**

17. As reviewer J7FP suggested, we have added the section"Partition Study" in the Appendix. J.**(Line1075-1114, Page20-21)**


We hope that our work provides valuable insights to the field of EEG-based BCIs by presenting a novel avenue for exploration: unsupervised individual continual learning designed for real-world scenarios.

**We sincerely welcome the reviewers' feedback and constructive insights to further refine and enhance our study.**

---

> ### Author Response · Authors · 2024-11-25
> **Rebuttal has been submitted and we are eager to hear your further constructive feedback**
>
> **Dear Reviewers (bzyV, BBh5, J7FP)**,
>
> We hope this message finds you well. We sincerely appreciate your valuable feedback on our paper. In response, we have made substantial revisions to address your concerns, including the following:
>
> - **Contribution of BrainUICL:** We have clarified our contributions to EEG-based applications and technological innovations.
> - **Related Work:** The section on Related Work has been reorganized, and we have incorporated recent studies to enhance its comprehensiveness.
> - **Baseline Comparison:** We have introduced a new comparative method based on EEG continual decoding.
> - **Data Preparation:** A detailed description of the data preparation process has been added.
> - **Additional Experiments:** We have included experiments that analyze performance variation within the training set, comparisons with other memory sampling methods, and a partition study.
> - **Technical Details:** We have supplemented the manuscript with detailed technical information regarding DCB and CEA.
>
> We kindly request your prompt review of our rebuttal to finalize the decision-making process. Your timely response is greatly appreciated.
>
> Thank you for your time and consideration.
>
> Best regards,
>
> The Authors

---

### Meta-Review · Area_Chair_QVdx · 2024-12-14

**Metareview:**

The submission presents a continual learning approach to EEG processing, evaluated on 3 datasets.  The submission received widely divergent reviews, with two very positive reviewers, one borderline, and one highly negative reviewer.  The negative reviewer had at least some factual inaccuracies in their review (comparability in previous reported settings, and the inclusion of patient data).  The authors seem to have satisfactorily rebutted the concrete concerns raised by the reviewer, and on the balance the submission is an interesting contribution to ICLR.

**Additional Comments On Reviewer Discussion:**

The discussion was highly active, and aside from one reviewer mentioned above, responses to the authors rebuttal were positive.

---

### Decision · Program_Chairs · 2025-01-22

Accept (Poster)